# The Determination of ClNO₂ via Thermal Dissociation-Tunable Infrared Laser Direct Absorption Spectroscopy

John W. Halfacre[1], Lewis Marden[1], Marvin D Shaw[1], Lucy J Carpenter[1], Emily Matthews[2], Thomas J. Bannan[2], Hugh Coe[2,3], Scott C. Herndon[4], Joseph R. Roscioli[4], Christoph Dyroff[4], Tara I. Yacovitch[4], Patrick R. Veres[5*], Michael Robinson[5,6], Steven S. Brown[5,7], Pete M. Edwards[1,8]

[1]Wolfson Atmospheric Chemistry Laboratories, Department of Chemistry, University of York, Heslington, York, YO10 5DD, UK

[2]Department of Earth and Environmental Science, Centre for Atmospheric Science, School of Natural Sciences, The University of Manchester, Manchester M13 9PL, UK

[3]National Centre for Atmospheric Science, University of Manchester, Manchester, UK

[4]Aerodyne Research, Inc., Billerica, MA, 01821, USA

[5]Chemical Sciences Laboratory, National Oceanic and Atmospheric Administration, Boulder, CO, 80305, USA

[6] Cooperative Institute for Research in Environmental Sciences, University of Colorado, Boulder, CO , 80305, USA

[7]Department of Chemistry, University of Colorado, Boulder, CO 80309, USA

[8]National Centre for Atmospheric Science, University of York, York, UK

*Now at National Science Foundation, National Center for Atmospheric Research, Boulder, CO, 80301, USA

*Correspondence to:* John Halfacre ([john.halfacre@york.ac.uk](mailto:john.halfacre@york.ac.uk)), Pete Edwards ([pete.edwards@york.ac.uk](mailto:pete.edwards@york.ac.uk))

**Abstract**. Nitryl chloride ($ClNO_2$) is a reservoir species of chlorine atoms and nitrogen oxides, both of which play important roles in atmospheric chemistry. To date, all ambient $ClNO_2$ observations have been obtained by chemical ionization mass spectrometry (CIMS). In this work, Thermal Dissociation Tunable Infrared Laser Differential Absorption Spectrometer (TD-TILDAS) is shown to be a viable method for quantifying $ClNO_2$ in laboratory and field settings. This technique relies on the thermal dissociation of $ClNO_2$ to create chlorine radicals, which undergo fast reactions with hydrocarbons to produce hydrogen chloride (HCl) that is detectable by the TILDAS instrument. Complete quantitative conversion of $ClNO_2$ to HCl was achieved at temperatures $> 400 °C$, achieving 1 Hz measurement precision of $11 \pm 1$ pptv ($3\sigma$ limits of detection of $34 \pm 2$ pptv) during laboratory comparisons with other $ClNO_2$ detection methods. After blank- and line loss-corrections, method accuracy is estimated to be within $\pm 5\%$. Performance metrics of TD-TILDAS during ambient sampling were a 1Hz precision of $19 \pm 1$ pptv and $3\sigma$ limits of detection of $57 \pm 3$ pptv), which is directly comparable to previously reported $ClNO_2$ detection by quadrupole CIMS. Thus, TD-TILDAS can provide an alternative analytical approach for a direct measurement of $ClNO_2$ that can complement existing datasets and future studies. The quantitative nature of TD-TILDAS also makes it a potentially useful tool for the calibration of CIMS instruments. However, interpretation of ambient data may be complicated by potential interferences from unaccounted-for sources of thermolabile chlorine, such as ClNO, chloramines, and organochlorides.

## 1 Introduction

Nitryl chloride ($ClNO_2$) is an important nighttime reservoir of two highly reactive atmospheric species: atomic Cl and $NO_2$. Atomic Cl radicals play multifaceted roles in oxidation chemistry throughout the boundary layer (Simpson et al., 2015),

including hydrocarbon oxidation (Atkinson et al., 2006a, and references therein), ozone production and destruction (Halfacre
and Simpson, 2022; Liao et al., 2014; Sarwar et al., 2012, 2014; Simon et al., 2009; Wang et al., 2016), and mercury depletion
(Driscoll et al., 2013). However, the quantitative magnitude to which they affect these processes remains an open question. On
the other hand, $NO_2$ is one of the principal components of photochemical smog and the major anthropogenic precursor for
ozone production. Accounting for all sources of $NO_2$ is therefore important for accurately informing chemical and air quality
models.

The first in situ observation of ambient $ClNO_2$ was reported by Osthoff et al. (2008) utilising chemical ionization

mass spectrometry (CIMS) in the polluted marine boundary layer. CIMS has since been used in a multitude of studies for
additional $ClNO_2$ observations worldwide, including North America (Jaeglé et al., 2018; Lee et al., 2018a, b; Mielke et al.,
2011; Riedel et al., 2012, 2013; Thornton et al., 2010; Wagner et al., 2013; Young et al., 2012), Europe (Bannan et al., 2015;
Phillips et al., 2012; Sommariva et al., 2018; Tan et al., 2022), Asia (Le Breton et al., 2018; Liu et al., 2017; Tham et al., 2016,
2018; Wang et al., 2022, 2016, 2017; Xia et al., 2020; Ye et al., 2021; Yu et al., 2020; Zhou et al., 2018), in the presence of
snow/ice (Kercher et al., 2009; McNamara et al., 2020), and in indoor air quality studies (Moravek et al., 2022). Limits of
detection are often reported at $10^0$ pptv under 25-30 s averaging times, (Bannan et al., 2015; Kercher et al., 2009; McNamara
et al., 2020; Mielke et al., 2011), and has been recently reported at sub-pptv for 1 s measurements (Decker et al., 2024). Typical
observed mixing ratios range from $10^1 - 10^3$ pptv, with the highest levels observed in coastal polluted regions, where sources
of nitrogen oxides and $Cl^-$-rich aerosols are plentiful (Wang et al., 2019, 2021, and references therein).

While CIMS is a highly effective technique, $ClNO_2$ quantitation involves non-trivial calibration work. A laboratory

source of $ClNO_2$ may be readily generated by flowing a known amount of $N_2O_5$ across a $Cl^-$-containing salt bed (or $Cl_2$ across
$NO_2^-$-containing salt bed), but its quantitation assumes unit conversion out of the salt bed (e.g., Osthoff et al., 2008) or requires
additional equipment to observe $ClNO_2$ thermal dissociation products, such as a thermal dissociation-cavity ring down
spectrometer (TR-CRDS) (Thaler et al., 2011) or a cavity attenuated phase shift spectrometer (CAPS) (e.g., Tan et al., 2022).
Further, $I^-$ based CIMS demonstrates variable sensitivities based on the temperature and relative humidity of the ion-molecule
reactor, thereby requiring substantial laboratory work to develop humidity- and temperature-dependent calibration factors (Lee
et al., 2014; Robinson et al., 2022). Thus, there is an opportunity to innovate a method that can detect $ClNO_2$ directly without
the need for supplemental instrumentation.

The advantages of optical methods include analyte specificity and near absolute detection, utilizing well-defined

physical absorption properties, and requiring only infrequent calibrations or method validation procedures. Thaler et al. (2011)
previously used a TR-CRDS system to detect $ClNO_2$ as $NO_2$ by absorption at 405 nm under laboratory conditions, achieving
CIMS-competitive metrics (e.g., reported 20 pptv limit of detection for 1 minute averaging). This was achieved by flowing
sample air through both an unheated reference pathway and a heated (450 °C) sample pathway, under which $ClNO_2$ would
thermally dissociate into Cl radicals and $NO_2$ (Reaction R1).

$ClNO_2 (g) + heat \rightarrow Cl(g) + NO_2(g)$                   (R1)

The difference in observed $NO_2$ signal between the two channels provided a quantitative $ClNO_2$ measurement. However, its
use for conducting field measurements was reported to be limited, as the thermal degradation of alkyl nitrates (i.e., PAN) into
$NO_2$ cannot be distinguished from $NO_2$ originating from $ClNO_2$ due to overlapping thermal dissociation profiles.

For this same thermal-dissociation setup, product chlorine radicals will react quickly (e.g., Cl radical lifetime of 0.2

s for typical $CH_4$ mixing ratios of 2 ppmv and $k_{298} = 1 \times 10^{-13}$ cm$^3$ molecule$^{-1}$ s$^{-1}$ (Bryukov et al., 2002)) with ambient
hydrocarbons (e.g., methane) to form hydrogen chloride (HCl), which is a stable reservoir species for reactive chlorine
(Reaction R2).

$Cl\ (g) + RH\ (g) \rightarrow HCl(g) + R(g)$ (R2)

Several optical methods for the high-frequency and precise detection of HCl have recently been reported that overcome
historical challenges with its sampling (Furlani et al., 2021; Hagen et al., 2014; Halfacre et al., 2023; Wilkerson et al., 2021),
making them attractive candidates for an alternative thermal dissociation approach for the detection of $ClNO_2$. In this work,
we demonstrate the coupling of a thermal dissociation furnace to HCl-TILDAS (TD-TILDAS) for quantitative detection of
$ClNO_2$ as HCl. Compared with CIMS, TD-TILDAS is a lower time-cost method for determining $ClNO_2$ mixing ratios,
involving less experimental calibration work and simpler data processing as a direct method.
## 2    Methods
### 2.1 $ClNO_2$ Generation
$ClNO_2$ was synthesized by flowing $Cl_2$ across a nitrite-rich slurry, as described by Thaler et al. (2011) and shown by Reaction
R3.

$Cl_2(g) + NO_2^-(aq) \leftrightarrow ClNO_2\ (g) + Cl^-(aq)$ (R3)

However, it is believed the $ClNO_2$, once produced, may react further by dissolving into the water, hydrolyzing, and producing
nitronium and chloride ions (R4) (Frenzel et al., 1998).

$ClNO_2\ (g) + H_2O\ (l) \leftrightarrow Cl^-\ (aq) + NO_2^+\ (aq)$ (R4)

The nitronium ion can then react with $NO_2^-$ to produce $N_2O_4$, which exists in equilibrium with $NO_2$ (R5).

$N_2O_4\ (g) \leftrightarrow NO_2\ (g) + NO_2\ (g)$ (R5)

As detailed by Thaler et al. (2011), this chemistry can be mitigated by minimizing the residence time of $ClNO_2$ in the reaction
vessel and, to a lesser extent, by increasing the $Cl^-$ content of the slurry to encourage the equilibrium in R4 towards $ClNO_2$.
Therefore, we composed our slurry using sodium chloride (>99.5% pure, BioXtra, Sigma Aldrich product no S7653-5KG,
USA) and sodium nitrite (99%, extra pure, Acros Organics Code 196620010, Belgium) at a mole ratio of 100:1 $Cl^-$:$NO_2^-$,
wetting with 18MΩ deionized water (Millipore). The slurry was housed in ~10 cm of 1.25 cm diameter PFA tubing. Varied
flow rates (0.5-5 mL min$^{-1}$) of 10 ppmv $Cl_2$ (diluted in nitrogen, BOC product no. 150916-AV-B, United Kingdom) were
injected into a dilution flow (ranging from 200-2499.5 mL min$^{-1}$) of $NO_x$-scrubbed compressed air (using trap composed of
50% Sofnofil (Molecular Products Ltd., Essex, United Kingdom) and 50% activated carbon) that was subsequently passed
over the slurry, generating $ClNO_2$. A portion of the dilution flow was directed into a bubbler containing 18MΩ deionized water
prior to entering the slurry to maintain a humid environment and prevent the slurry from drying out. A schematic diagram of
this setup is presented in Fig. 1a.
**2.2 TD-TILDAS**
The TILDAS instrument and operation technique have been well-described previously (McManus et al., 2011, 2015). HCl-
TILDAS was developed by Aerodyne Research, Inc. and characterized by Halfacre et al. (2023). Briefly, air is sampled at 3.0
L min$^{-1}$ through a heated (50 $^{\circ}$C) quartz "inertial inlet," which is a type of virtual impactor used to remove particles >300 nm
from the sample matrix. Sample air continues its flow through 3 m of heated (50 $^{\circ}$C) tubing into the Herriott cell (204 m
pathlength) inside the TILDAS. HCl is then detected via a mid-IR inter-band cascade laser that probes the strong R(1) H$^{35}$Cl
absorption line at 2925.89645 cm$^{-1}$ within the (1-0) rovibrational absorption band (Guelachvili et al., 1981).
Nitryl chloride was converted to HCl for detection by TILDAS via thermal dissociation and the subsequent reaction of Cl
radicals with hydrocarbons, namely methane (Reactions R1-R2) (Thaler et al., 2011). While modelling results predicted
ambient mixing ratios of methane (~2 ppmv) are sufficient for achieving unit conversion of 1 ppbv ClNO$_2$ to HCl (Sect. 3.1),
the sample flow was additionally spiked with propane (BOC Limited, product no. 34-A) to a mixing ratio of 5 ppmv to both
ensure reaction completeness and outcompete Cl wall losses, as the rate constant for the reaction between Cl and propane is 3
orders of magnitude faster than with methane (Atkinson et al., 2006a). Next, the sample was directed to a 90 cm length of
quartz tubing (9.5 mm OD, 7.5 mm ID) housed within a furnace (Carbolite Gero TS1 12/60/450) upstream of the inertial inlet.
Sixty centimetres of this tubing is held within the heated region of the furnace, resulting in a residence time of ~500 ms under
a flow rate of 3 L min$^{-1}$. The internal temperature of the furnace is monitored using the furnace's inbuilt temperature sensors
and logged using the furnace software. To mitigate HCl surface interactions after ClNO$_2$-conversion, perfluorobutane-1-
sulfonic acid (PFBS; Merck, product no. 562629, United Kingdom) vapor was introduced just after the furnace to actively
passivate tubing and inlet surfaces, improving HCl transmission to the TILDAS inlet by 1) displacing HCl sorbed to surfaces
and 2) increasing the non-polar character of surfaces by presenting a fluorinated chain to passing analytes (Halfacre et al.,
2023; Roscioli et al., 2016). As detailed by Halfacre et al. (2023), a flow (50-75 mL min$^{-1}$) of oxygen-free nitrogen was flowed
into the headspace of a Teflon bubbler containing 5 g of PFBS, thereby flushing the PFBS vapor into the sample line. A
schematic diagram of this setup is presented in Fig. 1b.
The major sources of uncertainty with using TD-TILDAS to detect ClNO$_2$ include the degree of ClNO$_2$ conversion
to HCl, instrument noise, background drifts, and potential line losses of HCl. Confirmation of the unit conversion of ClNO$_2$ to
HCl was confirmed by modelling and laboratory experiments (see Sects. 3.1 and 3.2). Instrument noise and background drifts
were assessed regularly from blanks. For laboratory experiments, blanks were performed by sampling the ClNO$_2$ standard
(Sect. 2.1) diluted in NO$_x$-scrubbed compressed air through the unheated furnace. This dilution air was generated using an air
compressor and dehumidifying system (dew point approximately $-60$ $^{\circ}$C, absolute water vapor concentration $\sim$ 0.01%). To
vary sample humidity, carrier gas flow was split such that varied amounts were passed through a bubbler containing deionized
water. Concerning line losses of HCl, the only source of HCl will be from ClNO$_2$ conversion during laboratory experiments,
and therefore line losses were assessed between the furnace and the inertial inlet. As detailed in Fig. 1b, 30 mL min$^{-1}$ of flow
from a homemade HCl permeation source (Furlani et al., 2021; Halfacre et al., 2023) was injected alternatingly before the
furnace and just before the inertial inlet to determine loss of HCl over this region. So long as unit conversion of ClNO$_2$ to HCl
can be confirmed and blank / line losses are corrected, this method will be as accurate as the TILDAS is for detecting HCl,
which was previously found to be within the 5% tolerance of a commercial HCl cylinder with a certified concentration
(Halfacre et al., 2023).
For ambient sampling (Fig. 1c), an additional 5m of 1.25 cm OD PTFA Teflon was added before the tee that splits
the CIMS and TILDAS flow paths to sample outside air. A 5 µm PFA Teflon filter was also installed to collect particulates,
reducing the potential for HCl displacement through thermodynamic partitioning of particulate Cl$^-$ that would otherwise enter

156 the heated furnace (Huffman et al., 2009). This Teflon filter was not found to affect the observed mixing ratio of our $ClNO_2$

157 standard in measurement comparisons with and without the filter. However, the collection of particulates on the filter could

158 enable heterogenous chemistry with passing $N_2O_5$ plumes that may produce corresponding $ClNO_2$ plumes that are not reflective

159 of ambient chemistry, and so frequent replacement of these filters is necessary (e.g., daily). Blank air was generated by

160 pumping ambient air through a 50% activated carbon / 50% Sofnofil scrubber, which was found to effectively remove $ClNO_2$

161 from the sample stream. The pump (KNF model N035.1.2AN.18) was able to overblow the sample inlet at a flow rate of ~25

162 L min$^{-1}$. This approach is favoured over the use of synthetic cylinder air as significant changes in sample humidity can result

163 in release of HCl from surfaces (Halfacre et al., 2023). Blanks were performed for 10 minutes every 30 minutes to ensure the

164 instrument had enough time to respond and adjust to a stable background value. Additionally, ambient measurements will

165 include HCl, which would act as an interference for $ClNO_2$ observations. To obviate this, a denuder (coating of 2% $Na_2CO_3$

166 and 2% glycerol dissolved in 50% water and 50% methanol) was installed before the furnace to selectively remove acidic

167 gases (e.g., HCl, $HNO_3$) that may influence quantitation. Using the denuder for this purpose was found to be effective for at

168 least one-week periods, after which it was generally replaced to avoid coating exhaustion. The denuder was also found to affect

169 $ClNO_2$ throughput on shorter term timescales (e.g., daily), with a freshly coated denuder causing as much as 55% loss of the

170 $ClNO_2$ standard mixing ratio. This was determined by calculating the percent difference when sampling the $ClNO_2$ standard

171 both through and bypassing the denuder. Because $ClNO_2$ additions during ambient sampling will always be added through the

172 denuder, it was important that the $ClNO_2$ standard (Sect. 2.1) was sampled in dry air before and after overnight experiments

173 to quantify how this loss evolved over the course of an experiment such that data could be corrected using the percent difference

174 term. Periodic additions of HCl standard were also performed to assess line losses of HCl after conversion in the furnace. In

175 contrast to the laboratory experiment configuration, permeation source HCl in blank air was only injected just downstream of

176 the furnace mid-experiment to reduce exposure of unpassivated sampling surfaces to HCl. Losses were assessed by comparing

177 this observed HCl injection value to pre- and post-experiment injections over dry compressed air. Injections of HCl and $ClNO_2$

178 standards was controlled using 3-way Teflon solenoid valves (MasterFlex Model no. 01540-18, Cole Parmer, United

179 Kingdom).

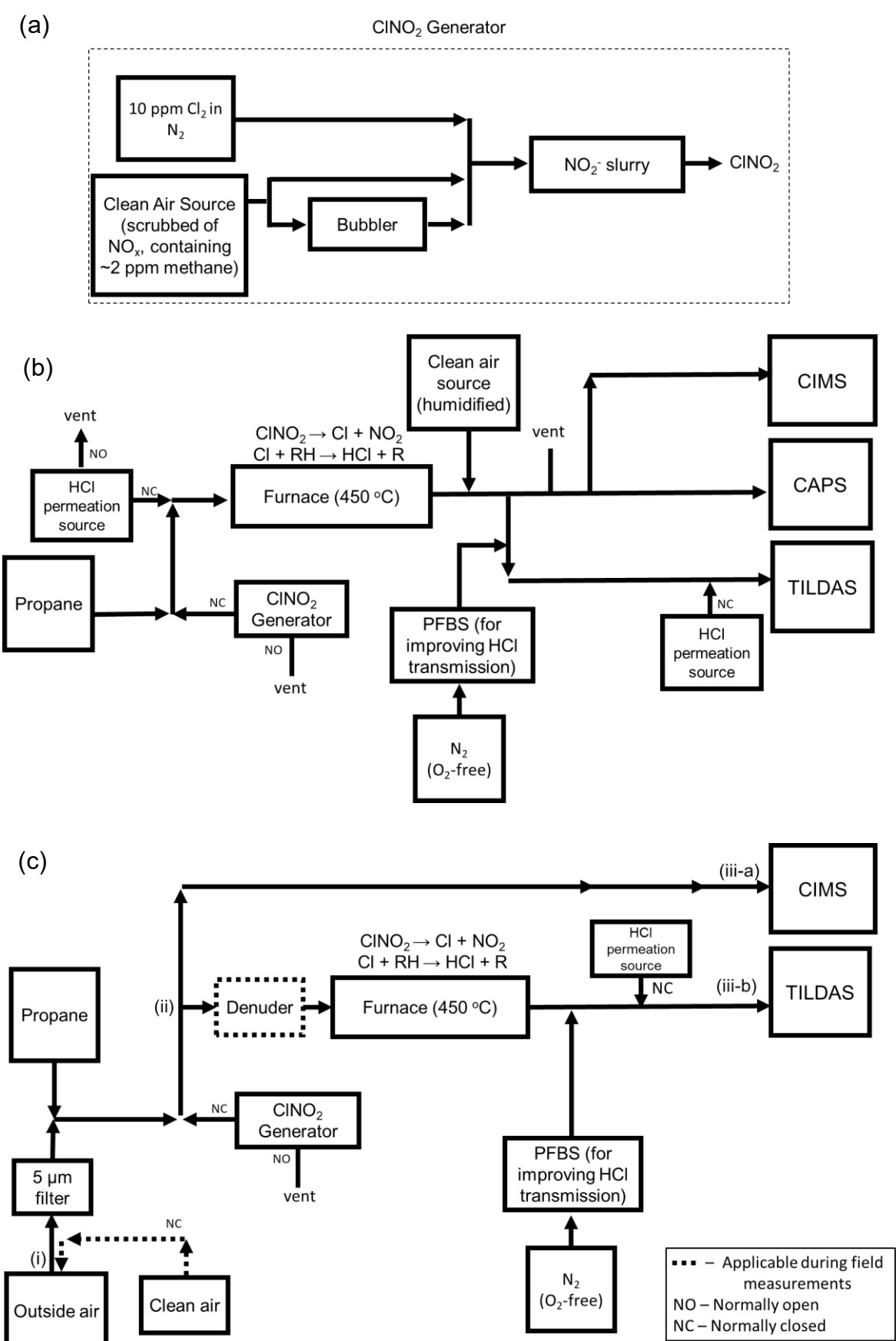

Figure 1 Experimental schematic diagrams for (a) generating ClNO₂, b) laboratory comparison measurements between CIMS, TILDAS, and CAPS NO₂, and (c) calibration/field sampling between CIMS and TILDAS. Note that "NO" stands for "normally open" and "NC" stands for "normally closed" in reference to solenoid valves that control the flow direction for these items. For (c), the approximate distance between point (i) and (ii) is 5 m, from (ii) to (iii-a) is 1.5 m, and from ii to (iii-b) is 2 m.

**2.3 Supporting Instrumentation**
To confirm the efficacy of TD-TILDAS as a valid quantitative method for $ClNO_2$ detection, testing was performed
simultaneously with a Cavity Attenuated Phase Shift (CAPS) $NO_2$ instrument (Sect. 2.3.1) and Time of Flight-Chemical
Ionization Mass Spectrometer (Sect. 2.3.2), both of which have previously reported as $ClNO_2$ detection methods.
**2.3.1 Cavity Attenuated Phase Shift (CAPS) $NO_2$**
$ClNO_2$ mixing ratios observed by the TILDAS were confirmed via simultaneous detection of the $NO_2$ product of $ClNO_2$ thermal
dissociation using a commercial Cavity Attenuated Phase Shift $NO_2$ detector (Teledyne T500U CAPS). Briefly, emission from
a LED (emission centred around 425 nm) is reflected across two spherical mirrors and absorbed by $NO_2$ in the optical cell.
This difference in light is detected by a photodiode and quantified based on its absorbance via the Beer-Lambert Law. The
instrument was calibrated using gas-phase titration of NO by $O_3$ to produce varied concentrations of $NO_2$. A 1ppm NO in
nitrogen cylinder (certified 982 ppb, NPL) was used to verify the concentration of NO in a 25ppm NO in nitrogen working
standard (BOC). A multigas blender (Environics S6100) was used to generate a range of $O_3$ concentrations (range 0-130 ppbv)
for titrating some of the NO (NO in excess, 200 ppbv) into $NO_2$, and the decrease in the NO concentrations was measured
using a calibrated $NO_x$ instrument (Teledyne API Chemiluminescence T200). The $NO_2$ introduced to the CAPS instrument is
thus the sum of the drop of NO from the added ozone and the $NO_2$ already present in the working standard. The T200 $NO_x$
instrument was also used to measure ambient air alongside the CAPS (range 0-25 ppbv), and these data are presented in Fig.
A1. Additionally, the Teledyne T500U includes an internal drying assembly and has a manufacturer recommended humidity
range of 0 – 95%.
**2.3.2 Time of Flight Chemical Ionisation Mass Spectrometry (CIMS)**
$ClNO_2$ was additionally detected using a VOCUS high-resolution chemical ionization time-of-flight CIMS (Tofwerk,
Switzerland) with a VOCUS AIM reactor and using iodide ($I^-$) as a reagent ion gas. A complete description of this instrument
and its operational principles are described in detail by Riva et al. (2024). Briefly, sample gas is drawn into the sampling inlet
and pulled through a critical orifice (0.475 mm) and PFA Teflon sample flow guide into a conical ion-molecule reactor (IMR)
at a flow rate of 1.8 L min$^{-1}$. The IMR was held at a constant pressure of 50 mbar using a vacuum pump (IDP3, Agilent
Technologies) and temperature controlled to 50 $^o$C. The reagent ion source was a permeation tube containing trace amounts of
$CH_3I$ dissolved in benzene (Tofwerk). Ultra-high purity, oxygen-free $N_2$ gas (generated by flowing compressed air through
gas with a commercial $N_2$ generator, Infinity NM32L, Peak Scientific Instruments, UK) is continually flowed over the
permeation tube to flush the gaseous $CH_3I$/benzene mixture into a compact vacuum ultraviolet ion source (VUV). Within the
VUV, UV light emitted from a Kr lamp (116.486 nm and 123.582 nm) is absorbed by benzene, generating low energy
photoelectrons that can react with $CH_3I$ to produce $I^-$ (Ji et al., 2020). The $I^-$ reacts with analytes for approximately 30 ms
before being drawn through another critical orifice where the sample travels through four differentially pumped chambers
before reaching the drift region of the ToF-CIMS. Ions in the ToF chamber are extracted and converted into mass spectra via
an MCP detector with a preamplifier over a mass range of 7-510 Th. The extracted packets are averaged over a period of 1
second and the resolution of the instrument is $\approx$ 5000. Data was collected at a rate of 1 Hz. Data averaging, mass calibration,
peak assignment, peak fitting and peak integration are all performed using the software package Tofware (version 4.0.0,
TOFWERK) used in Igor Pro 9 software (Wavemetrics). Peak fitting focused on $I^{35}ClNO_2^-$ (m/z 207.8668) and $I^{37}ClNO_2^-$ (m/z
209.8638), and isotope abundances were manually confirmed to be ~1:0.32, based on the natural abundance of chlorine
isotopes. CIMS signals were normalized against the sum of the total number of reagent ions, which is equivalent to $I^- + I(H_2O)^-$
. Additionally, as the CIMS sensitivity to $ClNO_2$ varies with humidity in the ion-molecule reactor region, we define an
additional term equal to ratio of the iodide water cluster ($I(H_2O)^-$) to the reagent ion sum ($I^- + I(H_2O)^-$), hereafter referred to as
the Iodide Water Ratio (IWR). Instrument backgrounds were assessed using air scrubbed of $ClNO_2$, as described in Sect. 2.2.
**2.4 Data Analysis**
Data analysis was conducted using the R language for statistical computing (R Core Team, 2021). Linear regressions were
calculated using the York method (Cantrell, 2008) when possible so as to incorporate uncertainties in compared variables.
**2.5 Chemical Modelling**
The 0-D box model Kintecus (Ianni, 2003, 2022) was used to explore the gas phase chemistry occurring in the heated furnace
to predict the timescales of the thermal-dissociation of $ClNO_2$ and the subsequent formation of HCl after reaction with
hydrocarbons (Reaction R2). The only hydrocarbon included in these model experiments was methane. The model was also
used to identify potential interferents that could prevent unit conversion of $ClNO_2$ to HCl. The results of the model were used
to guide the experimental set-up. The modelled species, reaction list, tested interferents (including ClNO and alkenes), and
initial concentrations are included in the Appendix (Tables A1-A3). Reaction kinetics were sourced from the NIST Chemical
Kinetics Database and IUPAC Evaluated Kinetic Data websites (Manion et al., 2015; Wallington et al., 2021), and primary
literature references are listed next to each reaction. No chemical species were held constant or were otherwise constrained
outside of initial concentrations. The model integration time was set to 1 ms, and the entire simulation was set to last 150 ms.
The model initiated with a temperature of 25 $^o$C (held for 10 ms) before increasing to 450 $^o$C over the course of 22 ms. The
temperature was held at 450 $^o$C for 40 ms, before gradually decaying back to 25 $^o$C over 70 ms.
**3 Results & Discussion**
**3.1 Modelling TD Chemistry**
Box model simulations predicted the rapid, virtually unit conversion of $ClNO_2$ to HCl after increasing temperature to 450 $^o$C
(Fig. 2) under the model conditions outlined in Tables A1-A3. Ninety percent conversion was calculated to occur within 23
ms from a starting $ClNO_2$ concentration of $2.46 \times 10^{10}$ molecules $cm^{-3}$ (1 ppbv at 25 $^o$C), and ambient mixing ratios of methane
(i.e., 2 ppmv at 25 $^o$C) were found to be sufficient for facilitating this chemistry. While Cl-mediated hydrocarbon oxidation
was shown to produce a modest enhancement of hydroxyl radical concentrations (Fig. 2b), it was not enough to compete
meaningfully with Cl to mitigate or retard Reaction R1. Similarly, an initial $O_3$ concentration of $9.84 \times 10^{11}$ molecules $cm^{-3}$
(40 ppbv at 25 $^o$C) did not significantly inhibit the desired chemistry by the direct reaction of $O_3$ with Cl radicals.

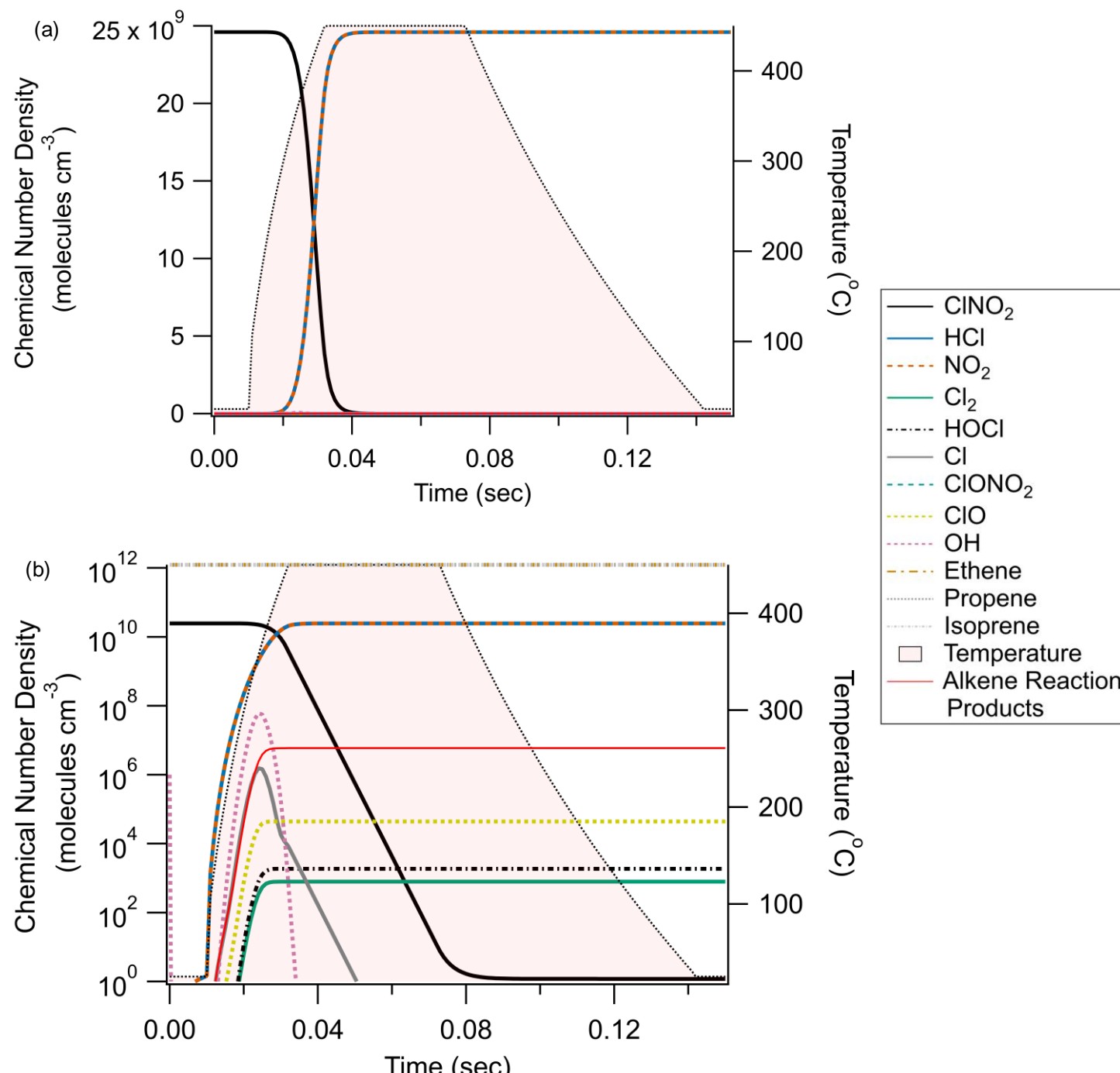

**Figure 2: Chemical modelling results of the thermal dissociation of ClNO$_2$ and its subsequent conversion to HCl. Panel A presents results on a linear y-axis, while Panel B features the same data on a logarithmic y-axis. Note that ethene, propene, and isoprene are off-scale in Panel A (1.23 x 10$^{12}$ molecules cm$^{-3}$ / ~50 ppbv) to better display the relationships between ClNO$_2$, HCl, and NO$_2$, and are shown to remain constant in Panel B.**



Concerning potential interferents, Cl can add to double bonds found on alkenes without producing HCl. Reactions with ethene,
propene, and isoprene were included in the model at 1.23 x 10$^{12}$ molecules cm$^{-3}$ (50 ppbv at 25$^o$ C) each and were found to
produce approximately 1 x 10$^6$ molecules cm$^{-3}$ of non-HCl product, which is 4 orders of magnitude less than the HCl converted
from ClNO$_2$. As these mixing ratios of alkenes are larger than those typically found in real world environments (e.g.,
Budisulistiorini et al., 2015; Hellén et al., 2024; Tripathi et al., 2021), it is therefore unlikely alkenes will cause meaningful
interference for $ClNO_2$ quantification.
$ClNO_2$ was predicted to be the only known inorganic chlorine reservoir to thermally dissociate at 450 °C. This is
consistent with the relative bond dissociation energies found for $ClNO_2$ (142 kJ $mol^{-1}$) relative to the various other forms of
inorganic chlorine simulated (Cl-$NO_2$ < Cl-Cl < Cl-O < Cl-R < Cl-H) (Darwent, 1970). Production of other inorganic chlorine
compounds (e.g., $Cl_2$, HOCl, $ClONO_2$, or reformation of $ClNO_2$) was orders of magnitude less than the resulting HCl and is
therefore not believed to influence HCl production. Even so, there remain potential inorganic chlorine species that may still
act as interferents in this method. Nitrosyl chloride (ClNO) has been previously predicted by modelling to exist at ppbv-level
mixing ratios in polluted marine environments and could be an efficient Cl-atom source (Raff et al., 2009). Indeed, 1 ppbv
(2.46 x $10^{10}$ molecules $cm^{-3}$) of ClNO was found to partially thermally dissociate in our Kintecus model (bond dissociation
energy of 159 kJ $mol^{-1}$) and generate additional HCl, as well as NO that was gradually converted to $NO_2$ (Fig. A2). On the
addition of heat, ClNO decreased by 40% while HCl increased by an equivalent amount (in addition to the 2.46 x $10^{10}$
molecules $cm^{-3}$ generated by $ClNO_2$ thermal dissociation). While we are unaware of any field measurements that have
confirmed the presence of ClNO in the boundary layer to date, it appears likely this method would be sensitive to interference
from ClNO if/where its presence is confirmed.
.Additionally, one notable class of compounds that could not be simulated were chloramines, which have recently
received increased attention as relevant daytime sources of Cl atoms (A. Angelucci et al., 2023; Wang et al., 2023). Their
largest known anthropogenic sources include water disinfection processes, swimming pools, and cleaning products.
Trichloramine, dichloramine, and monochloramine have reported bond dissociation energies of 381, 280, 251 kJ $mol^{-1}$,
respectively (Darwent, 1970) ($ClNO_2$ bond dissociation energy = 142 kJ $mol^{-1}$), and so would not be expected to produce free
Cl radicals in the temperature range simulated herein if its thermochemistry is consistent with the above bond dissociation
energy trend. However, to the authors' knowledge no information is available regarding their thermal stability in the gas phase
at atmospherically relevant conditions, and this potential source of positive interference for our proposed method cannot be
discounted via the model at this time. Similarly, prevalent organochlorides, such as methyl chloride ($CH_3Cl$), dichloromethane
($CH_2Cl_2$), chloroform ($CHCl_3$), and carbon tetrachloride ($CCl_4$) could cause positive interference if they dissociate and produce
Cl atoms in the furnace (World Meteorological Organization, 2022). Global average mixing ratios for $CH_3Cl$, $CH_2Cl_2$, $CHCl_3$,
and $CCl_4$ were ~ 550 pptv, ~40 pptv, 9 pptv, and 77 pptv, respectively, during 2020. Appropriate thermal dissociation kinetic
parameters could not be sourced for the conditions used herein (i.e, temperatures ≤ 450 °C), and so these compounds could not
be properly simulated by the Kintecus model. Similarly to the chloramines, the bond dissociation energies are much higher
than other compounds simulated (339, 310, 346, 293 kJ $mol^{-1}$ for $CH_3Cl$, $CH_2Cl_2$, $CHCl_3$, and $CCl_4$, respectively (Darwent,
1970; Weissman and Benson, 1983)).

## 3.2 Laboratory Characterization of TD-TILDAS

For laboratory characterization, a stable source of $ClNO_2$ was generated (Sect. 2.1) for assessing TD-TILDAS
performance in comparison with other established $ClNO_2$ sampling techniques, including CAPS $NO_2$ and CIMS (Sect. 2.3).
One key change between model simulations and this experimental setup is the inclusion of propane to the sample stream
(estimated mixing ratio of 5 ppmv within the heated section of sample configuration). While the model predicted the pertinent
chemistry will occur in ~23 ms using only ambient methane as the hydrocarbon (Sect. 3.1) and the residence time in the heated
furnace is ~500 ms, adding propane ensures complete conversion of $ClNO_2$ to HCl and ensures wall losses are negligible, as
Cl radicals react with propane approximately 3 orders of magnitude faster than with methane (Atkinson et al., 2006a). The fact
that no additional HCl signal was observed on addition of propane at varying levels (not shown) supports our calculations that
unit conversion is achieved and competitive loss of Cl radicals to walls is negligible.
A schematic diagram of these experiments is shown in Fig. 1a. Figure 3 represents a typical comparison experiment
in which $ClNO_2$ was sampled by all three instruments simultaneously. First, $ClNO_2$ was introduced into the flow stream with
the furnace unheated, yielding a positive CIMS signal for $IClNO_2^-$ (~1100 ncps for the example in Fig. 3), while TILDAS HCl
and CAPS $NO_2$ mixing ratios remained at background levels. As the furnace temperature approached 450 $^o$C, Reactions R1-
R2 began to occur. HCl and $NO_2$ mixing ratios rose, plateauing at similar values (~2.2 ppbv in Fig. 3) while $IClNO_2^-$ decreased
to the instrument baseline, implying both Reactions R1-R2 proceeded to completion. Signals returned to their original positions
once the furnace was allowed to cool back to room temperature (e.g., from 16:45 in Fig. 3). Note that HCl signal spike during
the furnace's temperature ramp was seen consistently across experiments, and was most likely caused by a shift in HCl
molecule partitioning between the surface of the quartz tubing toward the gas phase (Halfacre et al., 2023). Allan-Werle
deviation calculations demonstrate favourable performance metrics for TILDAS while sampling $ClNO_2$, with 1 Hz precision
of 11.8 pptv, and as good as 1.2 pptv with an integration time of 96 seconds (Fig. 4).
A summary of comparison experiments across varied humidities is presented in Fig. 5. The changes in HCl as
observed by TILDAS correlated strongly with the changes in $NO_2$ observed by the CAPS instrument (Pearson correlation
coefficients of 0.999, 0.997, and 0.987 for relative humidities of 11%, 44%, and 66%, respectively). However, the slopes were
consistently less than unity ($0.95 \pm 0.01$, $0.93 \pm 0.02$, and $0.91 \pm 0.02$ at 11%, 44%, and 66%, respectively), indicating observed
HCl mixing ratios were less than corresponding $NO_2$ mixing ratios. One potential explanation for this could be loss of Cl
radicals in the furnace, but we do not believe this to be the case (as detailed above). While physical losses of HCl to sampling
lines would not be unexpected as HCl has a high affinity for sorbing to physical surfaces, experiments were designed to
minimize these interactions, and line loss experiments were performed to quantify any losses observed at tested humidities.
Experimentally, a small flow (50-75 mL min$^{-1}$) of PFBS vapour was injected into the TILDAS sampling line downstream of
the furnace to reduce HCl affinity for surfaces (Sect. 2.2) (Note that PFBS was not introduced to the entirety of the flow path
to avoid sampling of PFBS by other instruments. Additionally, there is evidence that PFBS degrades at temperatures above
400$^o$C (Xiao et al., 2020), and so its ultimately efficacy and reproducibility within the furnace system would be uncertain).
Further, the high operating temperature of the furnace would also be expected to minimize HCl-wall interactions within the
quartz tubing. Indeed, no lines losses were found at 11% relative humidity between when the HCl permeation source standard
was injected into the sampling line before the heated furnace ($2.95 \pm 0.02$ ppbv) and when HCl was injected just before the
inertial inlet (accounting for dilution factors) ($2.95 \pm 0.02$ ppbv), consistent with Halfacre et al. (2023). Similar results were
found at 44% relative humidity (pre-furnace value of $2.68 \pm 0.03$ ppbv vs $2.66 \pm 0.03$ ppbv when HCl was introduced at inlet),
and real HCl loss was quantified at 66% relative humidity (pre-furnace value of $1.87 \pm 0.03$ ppbv vs $1.97 \pm 0.03$ ppbv when
HCl introduced at inlet). Having accounted for these line losses, ANOVA calculations found no significant differences between
these three slopes as presented in Fig. 3 ($F(2,19) = 0.10$, $p = 0.902$), indicating consistent performance between TILDAS and
CAPS for detecting $ClNO_2$. However, it does not appear to explain the deviation from unity, which will be discussed below.
As discussed in Sect. 2.1, chemistry may occur within the slurry to produce $N_2O_4$, which can easily degrade at room
temperature to produce two $NO_2$ molecules. If the $N_2O_4$ output from the $NO_2^-/Cl^-$ slurry is constant over the timescale of an
experiment (< 1 hr), it would be expected this additional $NO_2$ is readily accounted for during blank subtraction calculations.
While we believe this is largely true for the experiments presented above, discrepancies in $ClNO_2$ signals were observed as
the slurry aged (> ~3 weeks), with CAPS-observed $NO_2$ mixing ratios growing in significant excess of TILDAS-observed HCl
mixing ratios (Fig. A3). Separate applications of TILDAS- and CAPS-based calibration factors (using data from Fig. 5) to
concurrent CIMS $ClNO_2$ observations show closer resemblance to the TILDAS-observed mixing ratios (Fig. A3), suggesting

additional chemistry may be occurring within the salt bed that produces stable reservoirs of $NO_2$ that thermally dissociate in the furnace to produce undesired $NO_2$. This $NO_2$ artefact serves as a likely explanation for the sub-unity slopes presented in Fig. 5, as it would positively bias the CAPS measurements but not the TILDAS, which is only sensitive to HCl. Thaler et al. (2011) present in great detail strategies for minimizing $N_2O_4$ production in their study by minimizing the residence time in their $ClNO_2$ generator (0.3 s herein) and adjusting the molar ratio of $Cl^-:NO_2^-$ of their salt bed (100:1 herein), but were ultimately unable to completely eliminate it; while we found these strategies helpful for reducing the overall $NO_2$ background as measured by CAPS, we found they were unsuccessful in eliminating the artefact when sample gas was passed through the heated furnace. We are not aware of such chemistry being addressed in the literature for this $ClNO_2$ generation method and do not propose potential reactions as it is outside the scope of this paper.

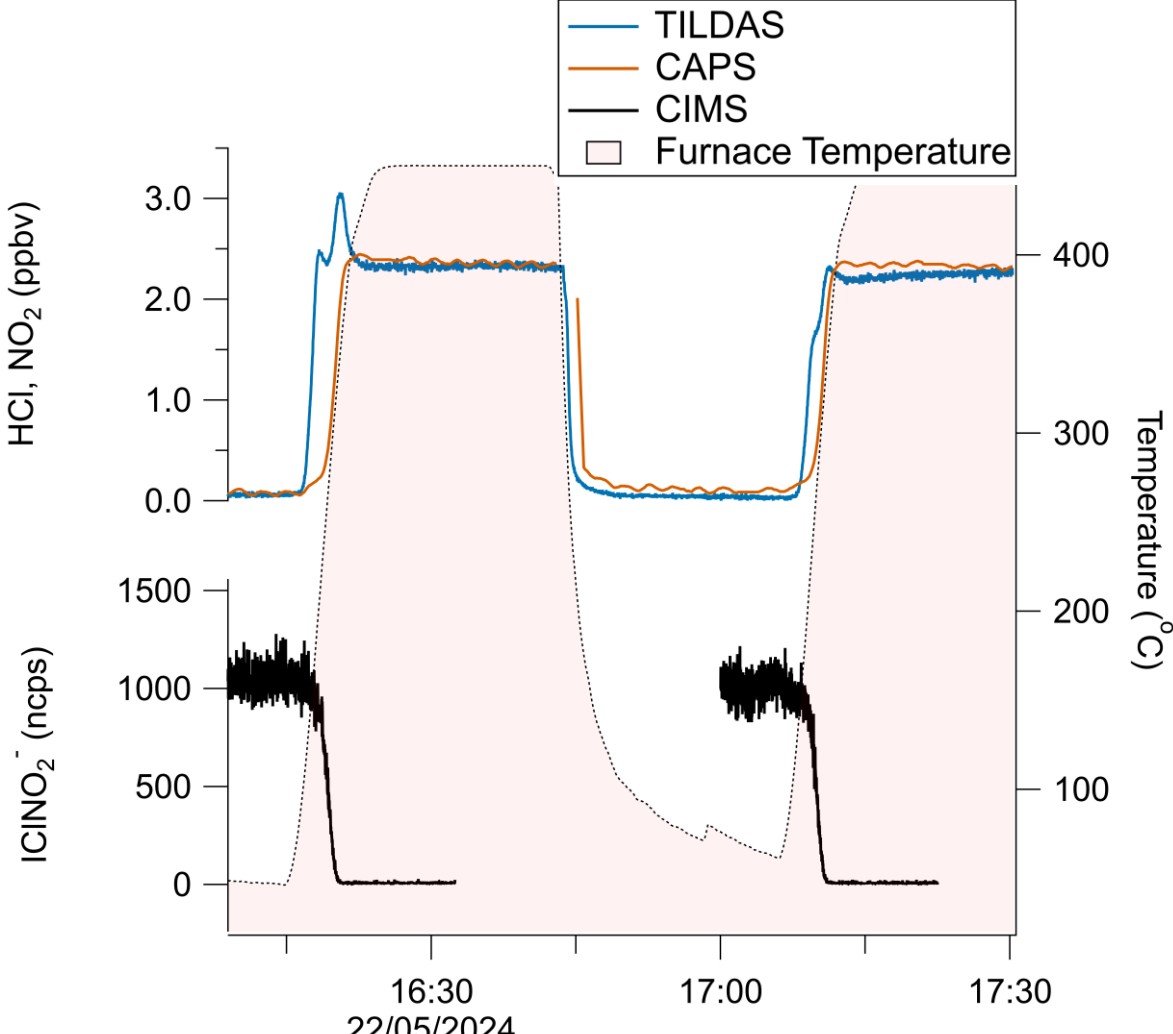

**Figure 3: a) Time series demonstrating the reversible thermal conversion of $ClNO_2$ to $NO_2$ (red trace, CAPS) and HCl (blue trace, TILDAS), as evidenced by changes in CIMS-observed $IClNO_2^-$ (black). Gaps in CIMS data are from internal CIMS tests not pertinent to this work.**

Both HCl and $NO_2$ mixing ratios independently correlated strongly with the CIMS measurement of $IClNO_2^-$ (Fig. 5b, c), and the $I^-$ CIMS sensitivity for $IClNO_2^-$ was found to vary strongly with humidity, as previously reported (Kercher et al., 2009; Mielke et al., 2011). The weakest Pearson correlation coefficient was for $NO_2$ and $IClNO_2^-$ at 66% relative humidity (r = 0.988), virtually matching that of $NO_2^-$ and HCl at the same humidity. Due to the uncertainty / unreliability of the $NO_2$ as it relates to $ClNO_2$ quantitation, we do not further consider the relationship between CAPS and CIMS.

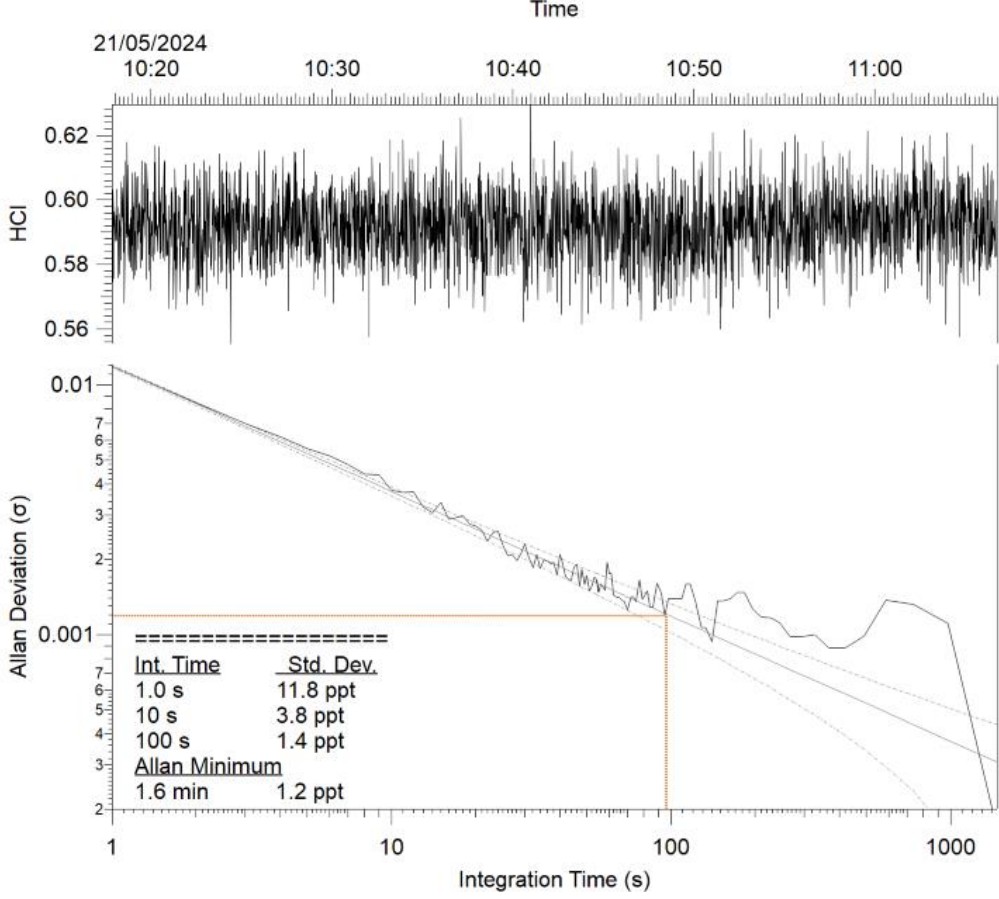

**Figure 4: Allan-Werle plot for TD-TILDAS during addition of ClNO₂ standard into the sample line. The Allan minimum is indicated by the dotted red lines.**

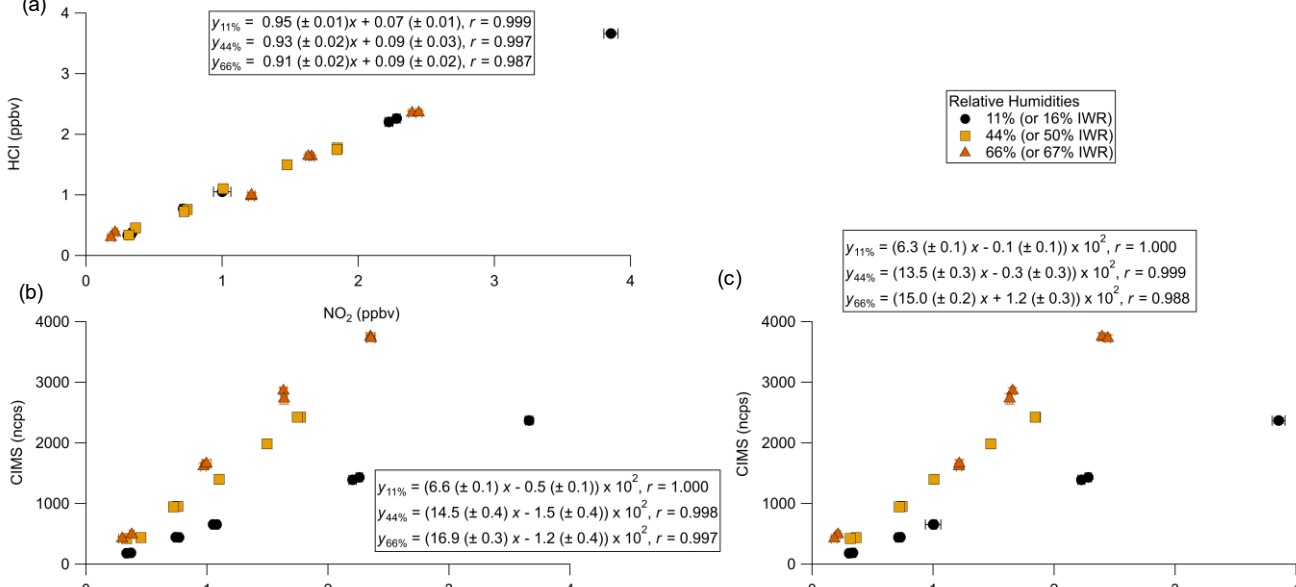

**Figure 5 – Comparison curves of a) TILDAS vs CAPS, b) CIMS vs TILDAS, and c) CIMS vs CAPS for injections of varied mixing ratios of ClNO₂ across different relative humidities. Regressions involving TILDAS data have been corrected for line losses observed at 66% relative humidity.**

The linear equations from Fig. 5a present significant intercepts that suggest a source of positive error for the TILDAS, and the similarity of these intercepts suggest a relatively constant/consistent source (values are statistically the same $F_{(4,19)} = 0.624$, $p = 0.546$ per ANOVA). For these experiments, TILDAS blanks were obtained by sampling slurry air flowed through an unheated furnace; in this scenario, Reactions R1-R2 are unable to occur, and therefore any signal observed by TILDAS could be considered background. It is possible that a small amount of HCl forms in the slurry system from the aqueous disproportion reaction between $Cl_2$ and $H_2O$. When the furnace is unheated, some amount of HCl interaction with the quartz tubing is expected given there is no PFBS flow through this portion of the plumbing, biasing this blank measurement low. Then, once the furnace is heated to 450 °C, this HCl will be liberated from the quartz tubing, possibly then biasing the heated measurement high. This is supported by the presence of a peak in observed HCl as the furnace reheats (e.g., as in the second temperature ramp in Fig. 3), as some of the HCl sorbed to the furnace tube walls under room temperature is forced into the gas phase. The statistical similarity in intercepts implies this effect is constant across these experiments, leading to a consistent offset. While an ideal blank would sample the gas downstream of the slurry while selectively scrubbing ClNO₂, this was not practical to achieve without simultaneously scrubbing HCl. Therefore, we propose the y-intercept in these cases is a good estimate of the systematic error present in these comparison experiments.

### 3.3 Applicability as Field Instrument

The applicability of TD-TILDAS as a field method for ClNO₂ detection was tested by sampling ambient air from outside the Wolfson Atmospheric Chemistry Laboratory building on the University of York campus (York, United Kingdom) from the morning of 13 January 2025 (Fig. 6). Compared with the laboratory-based configuration described in Sect. 3.2, ambient air will contain varied amounts of HCl that would interfere with accurate quantification of ClNO₂ via the TILDAS method. To address this, a base-coated denuder (Sect. 2.2) was installed in the HCl sampling line. ClNO₂ throughput was found to be hindered when flowed through the denuder but increased over the course of the observation period (pre-experiment estimation of 55% loss on 10 January vs 31% measured directly after the experiment on 13 January). This loss was accounted for by applying a time-varying, linearly interpolated correction factor for the denuder. In addition, line losses affecting HCl between the heated furnace and TILDAS inlet were estimated as 2.7%, which was added back into the TILDAS measurements. CIMS observations of $IClNO_2^-$ were calibrated against TD-TILDAS using a mid-experiment ClNO₂ addition, yielding a sensitivity factor of 1982 ncps ppb$^{-1}$ (measured with a corresponding IWR of 42%). We note that this factor is ~35% greater than the value of 1450 ncps ppbv$^{-1}$ as presented in Fig. 5b for a comparable IWR (44%); this is likely due to the replacement of the reagent ion permeation source, repair of reagent ion source heaters, and change in the overall sampling configuration between the experiments from Sect. 3.2 and this section (as illustrated by Fig. 1). Application of this sensitivity factor across this measurement period can be justified as the IWR was found to be stable (38 ± 2%). Limits of detection, based on instrument blanks, were found to average 10 ± 5 pptv for TD-TILDAS and 1 ± 1 pptv for CIMS (using 60 second data averaging).

As seen in Fig. 6a, TILDAS- and CIMS-observed ClNO₂ demonstrate very good agreement for these ambient observations in both signal magnitude and structure. This is quantitatively supported by regression calculations during this period that yield a slope of 0.97 ± 0.01 (Fig. 6b), which is well within the averaged combined uncertainty for this period of 9%. While the sub-unity slope could indicate small losses on the TILDAS method, pre- and post-experimental losses were tested and corrected for as detailed above, and so this is not believed to be a large source of error in this case. It is otherwise not unexpected that this slope is found to deviate from unity given the uncertainty in the application of a single-point CIMS

sensitivity factor. Nevertheless, this agreement gives us confidence that it is appropriate for these measurements and provides
proof-of-concept for this TILDAS method.

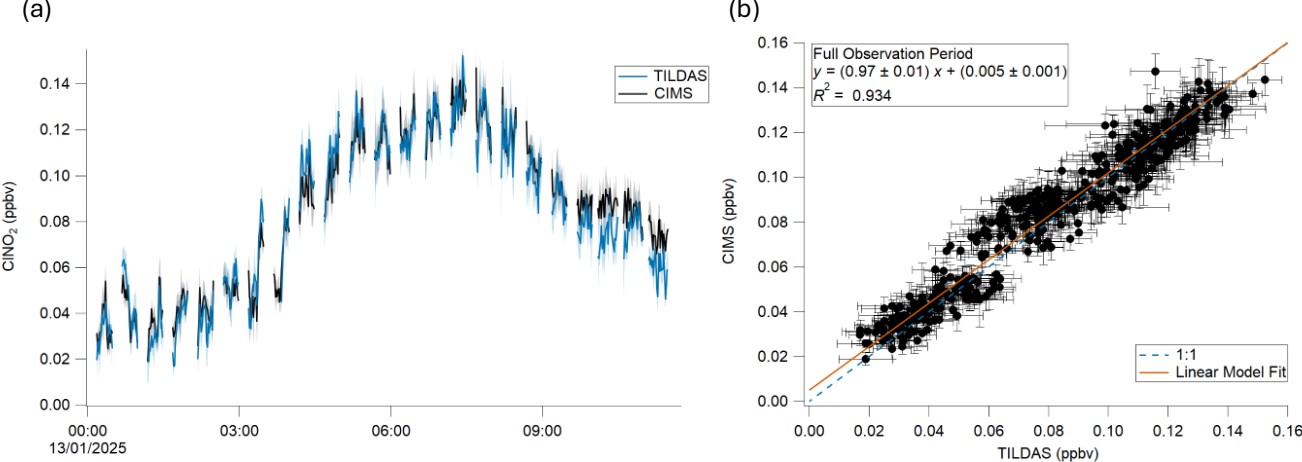


**Figure 6 – a) Time series comparison of TILDAS and CIMS observations of ClNO$_2$. b) Scatter plot of data shown in panel (a). The**
**error shading in (a) and bars in (b) represent the standard deviation of the 60 s averaged measurements.**

Additional sources of measurement uncertainty include unaccounted-for thermolabile chlorine reservoirs that could

cause positive interference in the TILDAS-method. As stated above, the TD-TILDAS method functions on the assumption
that ClNO$_2$ is the only major chlorine source that thermally dissociates at 450 °C. As shown by the model, ClNO may be a
potential source of interference if present (Fig. A2), while relevant thermochemistry information was unavailable for
organochlorides and chloramines, which therefore cannot be ruled out as possible interferences by modelling. Indeed, while
CIMS signals of ClNO and chloramines did not rise above their baselines during the period shown in Fig. 6, a separate
measurement period demonstrates multiple occurrences where signal increases in iodide-tri- and di-chloramine adducts (INCl$_3^-$
, INHCl$_2^-$) correspond with TILDAS-observed signal increases (Fig. A4). This is most dramatic at ~08:15, where ~115 ncps
of INCl$_3^-$ and 18 ncps of IHNCl$_2^-$ corresponds with an increase of 100 pptv in the TILDAS signal. While these chloramine
observations cannot be quantified at this time, trichloramine and dichloramine has previously been detected in downtown
Toronto at ≤ 0.104 ppb and ≤ 8 ppbv, respectively (Wang et al., 2023), suggesting a combined 100 pptv interference
contribution from these compounds is realistic. Synthesis and calibration of chloramine standards is a non-trivial task (Wang
et al., 2023), and so further experiments are required to investigate 1) to what extent the chloramine signals can be quantified
by TILDAS and 2) if the chloramine signal can be dissected from the ClNO$_2$ signal through temperature scans. The results of
such experiments may therefore allow this method to be extended for the quantification of both chloramines and ClNO$_2$.

While organochlorides (e.g., CH$_3$Cl, CH$_2$Cl$_2$, CHCl$_3$, and CCl$_4$) were not explicitly measured during the period in

Fig. 6, it would be expected that their potential interference in the TILDAS signal (if they dissociate in the furnace) would
present as a slow varying background signal that appears as an offset above a blank, given the ubiquity of these compounds
and relatively long tropospheric lifetimes for CH$_3$Cl, CH$_2$Cl$_2$, CHCl$_3$, and CCl$_4$ of 1 year, 6 months, 6 months, and 124 years,
respectively (World Meteorological Organization, 2022). Such an offset, if present, could be quantified during daytime
measurements (ie, when no ClNO$_2$ will be present in the boundary layer) and readily subtracted from nighttime measurements
if necessary. However, the agreement between TILDAS and CIMS measurements as presented in Fig. 6 suggests this
interference is not present, providing some evidence that these organochlorides are not dissociating in the furnace.
**Conclusions**
This work demonstrates the viability of TD-TILDAS as an independent $ClNO_2$ detection method at performance metrics
comparable to quadrupole CIMS, which are more than adequate for commonly observed mixing ratios in the boundary layer.
While modern CIMS instruments can achieve lower limits of detection and higher precision, the major advantage of TD-
TILDAS over CIMS is that it does not require external $ClNO_2$ calibration experiments, as this work demonstrates the unity
conversion of $ClNO_2$ to HCl that is subsequently detected based on well-understood spectroscopic principles. The TD method
described here can thus be used effectively in laboratory settings to measure $ClNO_2$ in related experiments, or even to calibrate
CIMS for $ClNO_2$ directly without needing to make assumptions regarding $Cl_2$ conversion on salt slurries. Additionally, use of
a denuder allows this method to be readily applied to other HCl optical instruments, such as those based on CRDS.

As a field method, TD-TILDAS demonstrated excellent agreement with a co-located CIMS for $ClNO_2$ detection. The

method is reliant on accurate and regular characterization of $ClNO_2$ throughput through the denuder, which was found to
increase across four days of sampling. Longer-term measurement campaigns would benefit from at least weekly denuder
replacements to ensure acidic gases are consistently scrubbed and do not interfere with $ClNO_2$ observations. However, the TD-
TILDAS method appears susceptible to positive interference, potentially resulting from chloramines, or other unaccounted-
for thermolysable chlorine compounds. Care should thus be taken should this method be deployed where large amounts of
chloramines are known to be present, such as swimming pools or near water treatment facilities. More work is still required to
confirm and quantify the response of this method to chloramine and organochlorides, and if so, identify an appropriate method
to mitigate this potential interference. While modelling additionally suggests ClNO as an interferent, its presence in the
boundary layer is yet to be confirmed through in situ observations. In any case, careful temperature ramps (e.g., Day et al.,
2002) performed with the furnace in environments where unknown interferences may be a concern would likely reveal the
purity of the $ClNO_2$ signal observed. Experimental adjustments could be further made for the TILDAS to alternate its sampling
between a heated channel (as described in this paper) for $ClNO_2$ detection and an unheated pathway that allows for the
additional detection of HCl. Doing so would require careful characterization of physical HCl losses inherent to both sampling
pathways, as well as consideration of the likely hysteresis in detected HCl mixing ratios resulting from changes to the sampled
air temperature that would affect the partitioning of HCl between surfaces and the gas phase.


**Appendix A**

**Table A1 – Bimolecular reactions and parameters used for the modelling described in Sect. 2.5. Reactions follow the rate expression $k(T) = A\,(T/298)^n\,e^{-Ea/RT}$ (Burkholder et al., 2015)**

| Reaction | A | n | Ea (kJ mol$^{-1}$) | Reference |
|---|---|---|---|---|
| $ClNO_2 + M \Longrightarrow Cl + NO_2$ | $9.13 \times 10^{-10}$ | 0 | 106 | (Baulch et al., 1981) |
| $Cl + Cl \Longrightarrow Cl_2$ | $6.15 \times 10^{-34}$ | 0 | -7.53 | (Baulch et al., 1981) |
| $M + ClONO_2 \Longrightarrow NO_2 + ClO$ | $2.76 \times 10^{-5}$ | 0 | 94.78 | (Anderson and Fahey, 1990) |
| $CH_4 + Cl \Longrightarrow CH_3 + HCl$ | $8.24 \times 10^{-13}$ | 2.49 | 5.06 | (Bryukov et al., 2002) |
| $HCl + OH \Longrightarrow H_2O + Cl$ | $3.74 \times 10^{-12}$ | 0 | 4.27 | (Baulch et al., 1981) |
| $HCl + M \Longrightarrow H + Cl$ | $7.31 \times 10^{-11}$ | 0 | 342 | (Baulch et al., 1981) |
| $CH_3 + HCl \Longrightarrow CH_4 + Cl$ | $3.89 \times 10^{-13}$ | 0 | 9.64 | (Baulch et al., 1981) |
| $CH_3 + NO_2 \Longrightarrow CH_3O + NO$ | $3.44 \times 10^{-11}$ | 0 | 0 | (Srinivasan et al., 2005) |
| $O_3 + M \Longrightarrow O + O_2$ | $7.6 \times 10^{-10}$ | 0 | 93.12 | (Heimerl and Coffee, 1979) |
| $CH_3 + O \Longrightarrow CH_2O + H$ | $2.26 \times 10^{-11}$ | 0 | 0 | (Baulch et al., 1992) |
| $HCl + O \Longrightarrow OH + Cl$ | $7.07 \times 10^{-14}$ | 2.87 | 14.72 | (Mahmud et al., 1990) |
| $OH + CH_4 \Longrightarrow CH_3 + H_2O$ | $4.16 \times 10^{-13}$ | 2.18 | 10.24 | (Srinivasan et al., 2005) |
| $Cl_2 + M \Longrightarrow Cl + Cl$ | $3.85 \times 10^{-11}$ | 0 | 196 | (Baulch et al., 1981) |
| $Cl + Cl \Longrightarrow Cl_2$ | $6.15 \times 10^{-34}$ | 0 | -7.53 | (Baulch et al., 1981) |
| $Cl_2 + O \Longrightarrow ClO + Cl$ | $4.17 \times 10^{-12}$ | 0 | 11.39 | (Baulch et al., 1981) |
| $Cl_2 + H \Longrightarrow HCl + Cl$ | $1.43 \times 10^{-10}$ | 0 | 4.91 | (Baulch et al., 1981) |
| $Cl_2 + OH \Longrightarrow HOCl + Cl$ | $3.60 \times 10^{-12}$ | 0 | 9.98 | (Atkinson et al., 2007) |
| $CH_3 + O_2 \Longrightarrow CH_3O + O$ | $2.19 \times 10^{-10}$ | 0 | 131 | (Baulch et al., 1992) |
| $ClO + O \Longrightarrow O_2 + Cl$ | $2.50 \times 10^{-11}$ | 0 | -0.91 | (Atkinson et al., 2007) |
| $OH + ClO \Longrightarrow HO_2 + Cl$ | $6.86 \times 10^{-12}$ | 0 | -2.49 | (Atkinson et al., 2007) |
| $OH + ClO \Longrightarrow HCl + O2$ | $4.38 \times 10^{-13}$ | 0 | -2.49 | (Atkinson et al., 2007) |

| Reaction | A | n | E | Reference |
|---|---|---|---|---|
| $CH_3O + NO ==> CH_2O + HNO$ | $4.00 \times 10^{-12}$ | -0.7 | 0 | (Atkinson et al., 1992) |
| $CH_3O + O_2 ==> CH_2O + HO_2$ | $7.20 \times 10^{-14}$ | 0 | 8.98 | (Atkinson et al., 1992) |
| $HOCl + O ==> OH + ClO$ | $1.70 \times 10^{-13}$ | 0 | 0 | (Atkinson et al., 2007) |
| $ClCO + M ==> CO + Cl$ | $4.10 \times 10^{-10}$ | 0 | 24.6 | (Atkinson et al., 2007) |
| $O_3 + NO ==> O_2 + NO_2$ | $1.40 \times 10^{-12}$ | 0 | 10.9 | (Atkinson et al., 2004) |
| $CH_3O_2 + NO ==> CH_3O + NO_2$ | $2.30 \times 10^{-12}$ | 0 | -2.99 | (Atkinson et al., 2006b) |
| $HO2 + NO ==> NO_2 + OH$ | $3.6 \times 10^{-12}$ | 0 | -2.24 | (Atkinson et al., 2004) |
| $CH_2O + Cl ==> HCl + HCO$ | $8.20 \times 10^{-11}$ | 0 | 0.28 | (Atkinson et al., 1992) |
| $CH_2O + OH ==> HCO + H_2O$ | $4.73 \times 10^{-12}$ | 1.18 | -1.87 | (Baulch et al., 1992) |
| $CH_3O_2 + HO_2 ==> CH_3OOH + O_2$ | $3.80 \times 10^{-13}$ | 0 | -6.49 | (Atkinson et al., 1992) |
| $CH_3OOH ==> CH_3O + OH$ | $6.00 \times 10^{14}$ | 0 | 177 | (Baulch et al., 1994) |
| $HCO + O_2 ==> CO + HO_2$ | $5.20 \times 10^{-12}$ | 0 | 0 | (Atkinson et al., 2006b) |
| $CO + OH ==> CO_2 + H$ | $5.40 \times 10^{-14}$ | 1.5 | -2.08 | (Baulch et al., 1992) |
| $Cl + HO_2 ==> HCl + O_2$ | $1.80 \times 10^{-11}$ | 0 | -1.41 | (Atkinson et al., 1992) |
| $Cl + HO_2 ==> ClO + OH$ | $6.30 \times 10^{-11}$ | 0 | 4.74 | (Atkinson et al., 2007) |
| $Cl + O_3 ==> ClO + O_2$ | $2.80 \times 10^{-11}$ | 0 | 2.08 | (Atkinson et al., 2007) |
| $CO + Cl ==> ClCO$ | $1.33 \times 10^{-33}$ | -3.8 | 0.00 | (Atkinson et al., 2007) |
| $OH + HOCl ==> H_2O + ClO$ | $5.00 \times 10^{-13}$ | 0 | 0 | (Atkinson et al., 2007) |
| $ClO + HO_2 ==> HOCl + O_2$ | $2.20 \times 10^{-12}$ | 0 | -2.8 | (Atkinson et al., 2007) |
| $ClO + ClO ==> Cl_2 + O_2$ | $1.00 \times 10^{-12}$ | 0 | 13.22 | (Atkinson et al., 2007) |
| $ClO + ClO ==> OClO + Cl$ | $3.50 \times 10^{-13}$ | 0 | 11.39 | (Atkinson et al., 2007) |
| $ClO + ClO ==> ClOO + Cl$ | $3.00 \times 10^{-11}$ | 0 | 20.37 | (Atkinson et al., 2007) |
| $ClO + NO ==> Cl + NO_2$ | $6.20 \times 10^{-12}$ | 0 | -2.45 | (Atkinson et al., 2007) |
| $CH_2O + O ==> HCO + OH$ | $1.78 \times 10^{-11}$ | 0.57 | 11.56 | (Baulch et al., 1992) |

| | | | | |
|---|---|---|---|---|
| OH + NO$_2$ ==> HNO$_3$ | 2.70 x 10$^{-11}$ | 0 | 0 | (Troe, 2012) |
| CH$_3$Cl + OH ==> CH$_2$Cl + H$_2$O | 1.40E-12 | 1.6 | 8.65 | (Cohen and Westberg, 1991) |
| CH$_3$Cl + H ==> CH$_3$ + HCl | 6.14E-11 | 0 | 38.9 | (Westenberg and deHaas, 1975) |
| CH$_3$Cl + CH$_3$ ==> CH$_4$ + CH$_2$Cl | 2.09E-12 | 0 | 48.6 | (Macken and Sidebottom, 1979) |
| CH$_3$Cl + Cl ==> CH$_2$Cl + HCl | 3.30E-11 | 0 | 10.39 | (Atkinson et al., 2008) |
| CHCl$_3$ + Cl ==> CCl$_3$ + HCl | 4.90E-12 | 0 | 10.31 | (Atkinson et al., 2008) |
| Cl + C$_3$H$_6$ ==> Products | 2.70E-10 | 0 | 0 | (Atkinson et al., 2006b) |
| Cl + C$_5$H$_8$ ==> Products | 4.30E-10 | 0 | 0 | (Orlando et al., 2003) |
| ClNO + M ==> Cl + NO | 2.16E-09 | 0 | 134 | (Baulch et al., 1981) |



**Table A2 – Termolecular reactions and parameters used for the modelling described in Sect. 2.5. The effective rate constant is**
**calculated by combing the low- and high-pressure limit expressions into the following formula:** $k_f(T, [M]) =$
$\left\{\frac{k_\infty(T)k_0(T)[M]}{k_\infty(T)+k_0(T)[M]}\right\} 0.6^{\{1+[log_{10}(\frac{k_0(T)[M]}{k_\infty(T)})]^2\}^{-1}}$

| Reaction | Low-Pressure Limit $k_0 = k_0^{298}(T/298)^{-n}$ | | High Pressure Limit $k_\infty = k_\infty^{298}(T/298)^{-n}$ | | Reference |
|---|---|---|---|---|---|
| | $k_0^{298}$ | n | $k_\infty^{298}$ | m | |
| Cl + NO$_2$ + M ==> ClNO$_2$ + M | $1.8 \times 10^{-31}$ | 2 | $1.1 \times 10^{-10}$ | 1 | (Burkholder et al., 2015) |
| CH$_3$ + O$_2$ + M ==> CH$_3$O$_2$ + M | $4.1 \times 10^{-31}$ | 3.6 | $1.2 \times 10^{-12}$ | -1.1 | (Burkholder et al., 2015) |
| Cl + C$_2$H$_4$ + M ==> Products | $1.6 \times 10^{-29}$ | 3.3 | $3.1 \times 10^{-10}$ | 1 | (Burkholder et al., 2015) |


**Table A3 – Initial concentrations for specified species simulated in model, and listed mixing ratios are based on a temperature of 20**
**°C. Potential interferents were tested in separate model runs according to the groupings on each line below, and were otherwise**
**initiated with a concentration of 0 molecules cm$^{-3}$. All other compounds were initialised with a concentration of 0 molecules cm$^{-3}$.**

| Species | Initial Concentration (molecules cm$^{-3}$) |
| --- | --- |
| $ClNO_2$ | $2.46 \times 10^{10}$ (1 ppbv) |
| $N_2$ | $1.92 \times 10^{19}$ (78%) |
| $O_2$ | $5.17 \times 10^{19}$ (21%) |
| $CH_4$ | $4.92 \times 10^{13}$ (2000 ppbv) |
| OH | $1 \times 10^6$ |
| $O_3$ | $9.84 \times 10^{11}$ (40 ppbv) |
| Potential Interferents | |
| ClNO | $2.46 \times 10^{10}$ (1 ppbv) |
| Ethene, Propene, Isoprene | $1.23 \times 10^{12}$ (50 ppbv) |




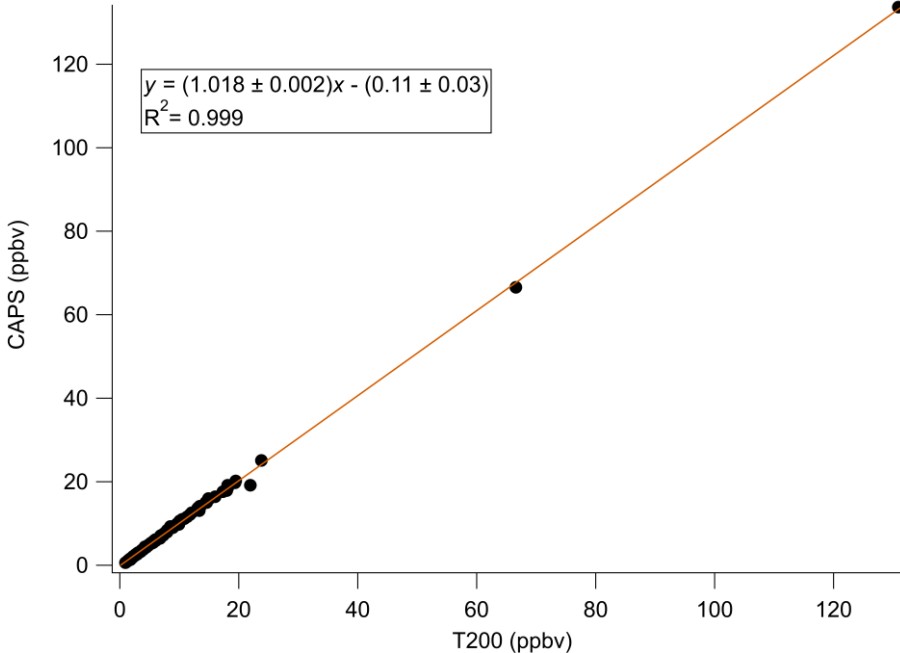


**Figure A1 – Laboratory calibration curve for CAPS NO$_2$**



(a)

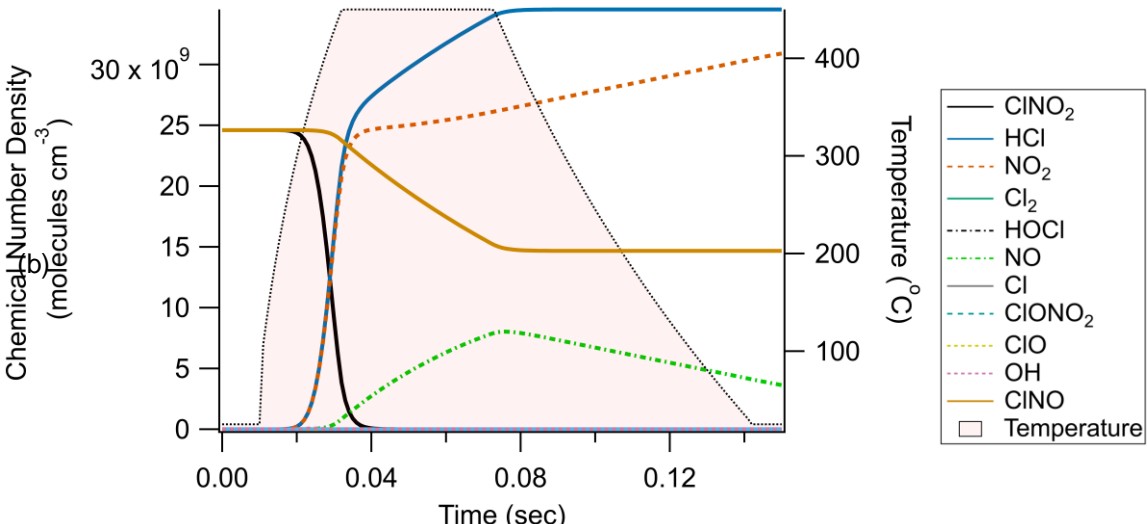


**Figure A2: Modelled effects of 2.46x $10^{10}$ (1 ppbv at 20°C) of ClNO.**



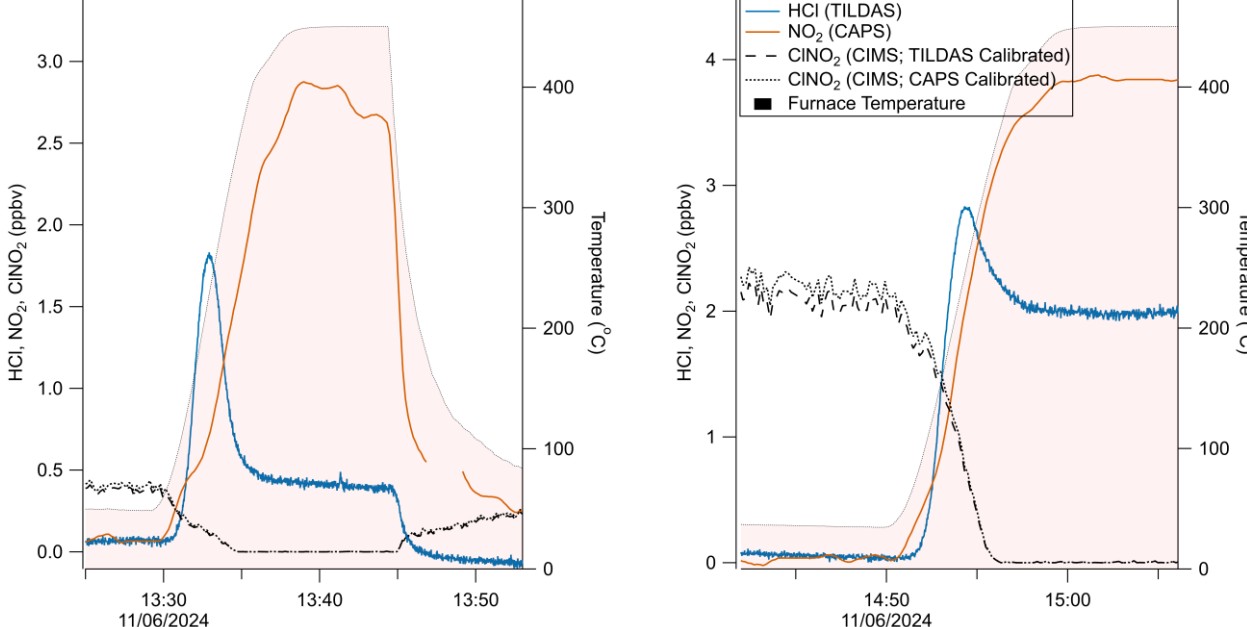


**Figure A3 – a) Comparison plot of ClNO₂ observations with an apparent excess of NO₂ formed after aging/processing of the same**
**slurry used for generating Fig. 3-4. (b) Additional comparison using a freshly made slurry. CIMS signal was calibrated using**
**humidity-dependent calibration factors as presented in Fig. 5.**


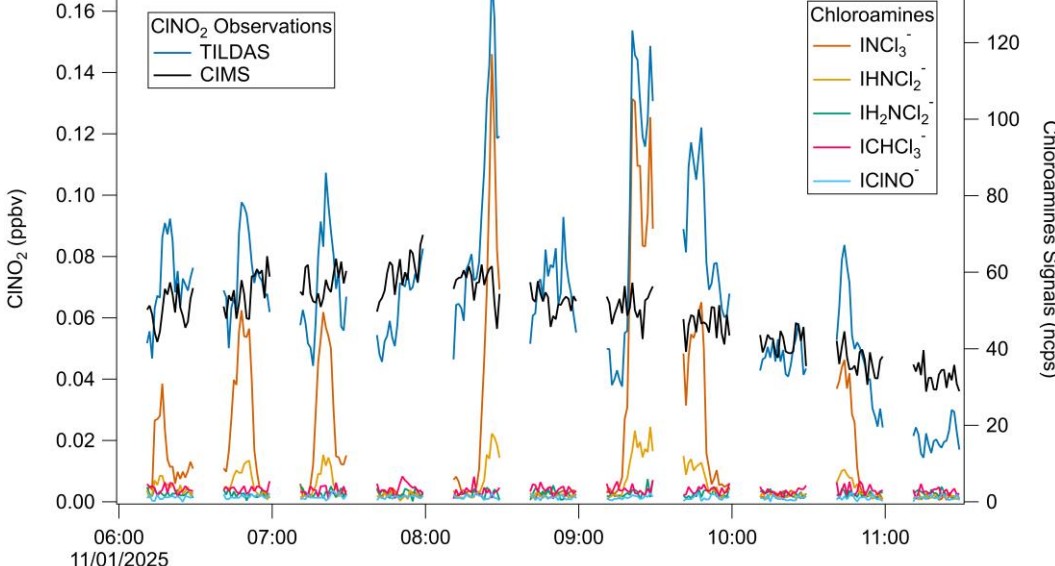


**Figure A4 –Field data showing apparent coincident signal increases TILDAS-observed ClNO₂ with CIMS-observed chloramines,**
**IClNO⁻, and ICHCl₃⁻ (uncalibrated).**

**Code availability**
Code used for this analysis is available from the corresponding author on request.
**Data availability**
Data are available from the corresponding author on request.
**Author contribution**
PRV, MAR, SSB designed and performed proof-of-concept experiments to demonstrate potential of the method.
SCH, HRR, CD, TIY designed, built, and tested the HCl TILDAS at Aerodyne Research, Inc. SCH, HRR, CD, TIY, and PME
designed initial laboratory experiments.
JWH and PME designed laboratory and field experiments, and JWH conducted laboratory and field experiments presented in
this work.
LM, MDS, LJC provided support for laboratory use of CIMS. EM, TJB, HC provided support for field CIMS observations.
JWH prepared the manuscript, and all authors reviewed the manuscript.

**Competing interests**
The authors declare that they have no conflicts of interest.
**Acknowledgements**
The authors would also like to thank Abigail Mortimer for her glassblowing services, Stephen Andrews and Stuart Young for
assistance with creating custom furnaces. Additionally, the authors thank William Drysdale and Katie Read for assistance with
calibrating and using the York CAPS instrument. Further, the authors thank Michael Agnese and Michael Moore for TILDAS
technical support.

**Financial Support**
This research has been supported by the European Research Council (H2020, grant no. ERC-StG 802685).

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
