# Peer review of "The Determination of ClNO2 via Thermal Dissociation-Tunable 2 Infrared Laser Direct Absorption Spectroscopy"

_EGUsphere, 2025_

## Author Comment (AC2)

We thank all three Anonymous Referees for their careful readings of our manuscript and appreciate their thoughtful feedback. We have edited the manuscript to incorporate their critiques, which we believe improves the overall quality of this work.

Below, individual Referee comments are presented in bold, and our responses are written in regular type. Line numbers in responses refer to the revised, "track changes" version of the manuscript, which will be uploaded separately. Altered figures will be replaced in both the "track changes" manuscript and revised manuscript, but changes to figures will only be described in the responses to the Referee below.

**Anonymous Referee 1**

**Some technical clarifications are needed in Section 2.2: the conversion to HCl is done using a flow of methane and propane, is that right? If so, in which proportions? Please amend Figure 1 to include methane, and briefly explain why two different VOC are needed.**

The primary hydrocarbons involved in HCl generation from $ClNO_2$ in this method are methane and propane, as the Referee correctly inferred. No additional methane is added from what is already present in ambient air and our zero-air source, and so the proportion of hydrocarbons available for reaction will be ~2 ppmv methane from ambient air and 5 ppmv propane from our cylinder. We have amended Fig. 1a to clarify that the air source for the $ClNO_2$ generator contains methane. Further, while the Kintecus modelling (Sect. 3.1) suggests ambient amounts of methane on its own should ensure complete reaction of 2 ppbv of $ClNO_2$ to HCl, the stream of propane is added to ensure possible wall loss of Cl and reaction of Cl with alkenes is negligible. Propane was chosen because its rate constant for its reaction with Cl radicals is 3 orders of magnitude greater than that with methane. We have clarified these points on lines 125-129.

**When sampling ambient air could there be interference from ambient VOCs? Specifically, VOCs with double C bond which would not form HCl. Can you comment on this point?**

The Referee is correct that alkenes, when present in the furnace, are likely to react with generated Cl atoms. However, this chemistry is not anticipated to affect this method to a significant extent given that we supply additional alkane (ie, 5 ppmv propane) to encourage complete conversion of Cl atoms to HCl. To demonstrate this, we have performed additional model simulations with Kintecus, adding 50 ppbv each of ethene, propene, and isoprene (i.e., amounts larger than those found in polluted environments, (e.g., Budisulistiorini et al., 2015; Hellén et al., 2024; Tripathi et al., 2021)). This was estimated to produce approximately $1 \times 10^6$ molecules $cm^{-3}$ of non-HCl product, which is more than four orders of magnitude lower than the resulting HCl. We therefore conclude that double bonded VOCs at typical mixing ratios are unlikely to cause a significant interference for this method.

We have amended Fig. 2 to include double bond products and now include these reactions in the Appendix Tables 1-2, as well as listed our starting concentrations in Table 3. We have amended the Fig caption accordingly. We have additionally added text describing the model testing of these interferents to Sect. 3.1 (lines 255 – 260).

**It is not clear what the procedure to account for ClNO2 loss in the denuder is: a 55% loss is quite high and it isn't constant with time (page 13). Are the data corrected for this loss in post-processing? If so (section 3.3 suggests as much), how reliable/stable this correction is? With regard to the denuder, is its efficiency in removing HCl constant with time?**

The Referee is correct that the data are corrected during post-processing. The procedure for denuder-related loss characterization is to measure $ClNO_2$ signal on TILDAS when injecting the $ClNO_2$ standard through the denuder, then taking a similar measurement with the denuder bypassed, followed by calculating the percent difference, which is ultimately added back into the observation. We have added language to this effect in Sect. 2.2 lines 168-175. The evolving $ClNO_2$ throughput through the denuder in between these characterizations suggest that the correction is only valid for the period analyzed, and characterization of other periods would necessitate explicit re-characterization of the denuder.

With regard to the denuder's complete removal of HCl, our experience from separate HCl studies finds the denuder to be effective on the order of 1-2 weeks when used as a periodic blanking mechanism and comparable to air scrubbed through a $CO_3^{2-}$ doped glass wool trap. In these cases, air is sampled through the denuder for approximately 2 minutes every 10 minutes, compared to this study where air is constantly flowed through the denuder. Anecdotally, we have found it is obvious for both $ClNO_2$ and HCl sampling when the denuder is exhausted, as the HCl signal through the denuder becomes erratic / excessively noisy in comparison with a freshly coated denuder, such as in the below figure:

[Figure]

The TILDAS valve-state values are included to more clearly indicate when the instrument sampling state is changing (0 = ambient sampling, 2 = blank, 16 = addition of ClNO2 standard). During this $ClNO_2$ ambient sampling configuration, TILDAS-sampled air is constantly passing through the denuder in both the blank and ambient sampling modes. When the denuder is exhausted, the signal becomes obviously poor (e.g., ~09:10). After replacement of denuder ~11:00, it can be seen that measurement precision is restored. We have added a statement to state we have found the acidic scrubbing nature of the denuder to be effective for at least one week on lines

168-169, and added recommendation for regular denuder replacement in the Conclusions section (Lines 429 – 432).

**Can you clarify the use of the HCl permeation source in the ambient setup? Lines 163-164 are not very clear. Why not leave the permeation source in front of the furnace (after the denuder). Also, briefly explain how PFBS mitigates HCl surface losses.**

On the role of HCl permeation source - The HCl permeation source was used in ambient experiments to quantify line loss in between the furnace and the TILDAS inlet. While the heated air coming out from the furnace and addition of PFBS will lessen HCl sorption to the line, our previous work (Halfacre et al., 2024) demonstrates that relative humidities above 60% can still cause ~6% permanent loss, and so it is important to perform regular additions of HCl standards to ensure the magnitude of this loss is quantified. We have added text to make this point clearer, lines 175-177.

On moving the permeation source downstream of the furnace – The permeation source was moved for ambient experiments to reduce the amount of unheated / unpassivated surface area on instrument plumbing exposed to HCl. Introducing HCl to untreated surfaces is likely to lengthen the time HCl takes to reach the detector due to interaction with these surfaces. While the experiments in Sect. 3.2 demonstrated negligible losses in the laboratory experiment configuration at lower relative humidities, this loss effect would be exacerbated at the relative humidities likely experienced in ambient sampling. As HCl-derived from $ClNO_2$ would only otherwise be created in the furnace in the ambient configuration (as ambient HCl is scrubbed by the in-line denuder), it is therefore only necessary to assess HCl losses downstream of the furnace. This is now clarified, lines 177-178.

On using PFBS to mitigate HCl interaction with surfaces – Roscioli et al (2016) reported the use of PFBS to increase acidic gas transmission through their instrument inlets. It is believed that 1) the strong sulfonic acid group displaces sorbed HCl (or other acidic gas) on surfaces, and 2) the sorbed PFBS presents a highly non-polar, fluorinated chain to passing gases, such that polar analytes will not interact with the coated surface. We have added this description to Sect. 2.2, lines 133-137.

**In section 3.2. The authors attribute the discrepancy between the TILDAS and the CAPS to potential NO2 artefacts from the salt bed. What is the residence time in the reactor? Did you attempt to increase it or reduce it, to see if that would affect the artefact? Did you vary the composition of the salt bed and/or its water content? I agree that speculation on the exact mechanism is out of place in this paper, but if these tests were done a brief comment would be useful.**

Much work was done to reduce the $NO_2$ artefact measured by CAPS, most of it unsuccessful. We found both the $NO_2$ background and artefact (when passed through the heated furnace) was greatest when using only $NO_2^-$ in the salt bed (i.e., no added $Cl^-$), and when using low flow rates (on the order of $10^2$ mL min$^{-1}$), leading to longer residence times in the $ClNO_2$ generation reactor. We found that both the $NO_2$ background and artefact were smallest when minimizing the residence time through the generator and using a 100:1 mole ratio of $Cl^- / NO_2^-$, consistent with Thaler et al, but we were ultimately not able to eliminate the artefact. We also found the artefact present both when the dilution air was dry or included added humidity. For the laboratory experiments in Sect. 3.2, the flow rate through the $ClNO_2$ generator was 2.5 L min$^{-1}$, resulting in a residence time of approximately 0.3

s. We have added clarifying text to indicate our experience is consistent with Thaler et al. (2011) and refer the reader to their thorough optimization work for more details (Sect. 3.2, lines 343-348).

**Chloramines are mentioned as potential interferences. Are there other compounds that could possibly create interference? Like Cl2, and chlorinated-VOCs?**

Where thermal dissociation data is available, the only inorganic forms of Cl known to dissociate at 400 °C are $ClNO_2$ and potentially ClNO (for which we do not know of any ambient measurements to date). This was consistent with the Kintecus model, in which starting mixing ratios of $Cl_2$, HCl, HOCl, and ClO were found to be unresponsive to the temperatures used in this study.

As it concerns chlorinated-VOCs, these compounds could potentially interfere if they dissociate at the temperature used in this study (i.e., 450 °C or 723 K). The UNEP Scientific Assessment of Ozone Depletion: 2022 includes trends in many organochlorides from ground sites between 1990-2020, including methyl chloride ($CH_3Cl$), dichloromethane ($CH_2Cl_2$), chloroform ($CHCl_3$), and carbon tetrachloride ($CCl_4$), which are four examples of widely observed chlorinated VOCs. Global average mixing ratios for $CH_3Cl$, $CH_2Cl_2$, $CHCl_3$, and $CCl_4$ were ~ 550 pptv, ~40 pptv, 9 pptv, and 77 pptv, respectively, during 2020, and so could introduce significant error if they do dissociate in our furnace.

Model investigation of the thermal dissociation of these compounds is complicated by scarce kinetic data at pertinent temperature ranges and/or use of noble gas-based bath gases instead of air. Using the NIST Kinetics Database (https://kinetics.nist.gov/kinetics/, accessed 12 May, 2025), rate constants for pertinent temperature ranges could only be found for the thermal dissociations of $CCl_4$ and $CHCl_3$ using theory-based studies :

| Reaction | A | n | Ea (kJ mol$^{-1}$) | Reference |
|---|---|---|---|---|
| $CCl_4 \rightarrow CCl_3 + Cl$ (300 – 800 K) | 9.13 x 10$^{-5}$ | 0 | 284 | (Huybrechts et al., 1996) |
| $CHCl_3 \rightarrow HCl + CCl_2$ (300 – 2000 K) | 1.74 x 10$^{19}$ | -8.7 | 267 | (Zhu and Bozzelli, 2003) |

When these kinetic from these studies are incorporated into our Kintecus model for 100 pptv of both compounds ($2.46 \times 10^9$ molecules cm$^{-3}$), dissociation for neither $CHCl_3$ nor $CCl_4$ was observed (purple and blue traces), and no additional HCl was formed:

[Figure]

The available experimental data otherwise cover temperature regimes greater than those used in our furnace, with noble gas-based bath gases, and/or under non-atmospheric pressure, and therefore may not properly represent the chemistry in the furnace. Nevertheless, we performed individual model runs that incorporated the following kinetic parameters:

| Reaction | A | n | Ea (kJ mol$^{-1}$) | Reference |
|---|---|---|---|---|
| $CCl_4 + M \rightarrow CCl_3 + Cl$ (773 – 873 K) | $4.59 \times 10^9$ | 0 | 189 | (Schiemann et al., 1962) |
| $CHCl_3 + M \rightarrow HCl + CCl_2$ (758 – 923 K) | $2.6 \times 10^{11}$ | 0 | 197 | (Shilov and Sabirova, 1957) |
| $CH_2Cl_2 + M \rightarrow CH_2Cl + Cl$ (1400 -2300 K) | $1.33 \times 10^{16}$ | 0 | 324 | (Lim and Michael, 1994) |
| $CH_3Cl + M \rightarrow CH_3 + Cl$ (1120-1150 K) | $1 \times 10^{15}$ | 0 | 358 | (Shilov and Sabirova, 1959) |

Using these data, the organochlorides completely and quickly dissociated at 450ºC (723 K) and created additional HCl:

[Figure]

Given the above contradictory results between the theoretical and experimental rate constants, we posit that the use of the available kinetic data for this purpose is highly uncertain, and that explicit experiments are required to determine whether these (or other) organochlorides dissociate in the furnace.

In any case, if it is found any of these compounds do in fact dissociate in our setup, their signal contribution to the TILDAS measurement would most likely be reflected by a slow varying background signal that appears as an offset above a blank, given the ubiquity of these compounds and the relatively long tropospheric lifetimes for $CH_3Cl$, $CH_2Cl_2$, $CHCl_3$, and $CCl_4$ of 1 year, 6 months, 6 months, and 124 years, respectively (per the UNEP Scientific Assessment of the Ozone Layer Depletion: 2022 report). This offset could be estimated during daytime measurements (ie, when no $ClNO_2$ will be present in the boundary layer) and readily subtracted from nighttime measurements if necessary. However, the excellent agreement between CIMS and TILDAS $ClNO_2$ as

presented in Fig. 6 of the manuscript would suggest no such offset was present during our field measurements, providing some evidence that these organochlorides are not dissociating in our furnace.

We have amended the manuscript to clarify that chlorinated VOCs could not be ruled out as interferences by the model as appropriate kinetic parameters were scarce (lines 284-291). We have also added how the presence of this interference would be detected and accounted for (lines 424-431), and suggested in the Conclusions that users of this technique perform temperature ramps with the furnace in environments in which the user is concerned of interferences (lines 450-452)

References:

Budisulistiorini, S. H., Li, X., Bairai, S. T., Renfro, J., Liu, Y., Liu, Y. J., McKinney, K. A., Martin, S. T., McNeill, V. F., Pye, H. O. T., Nenes, A., Neff, M. E., Stone, E. A., Mueller, S., Knote, C., Shaw, S. L., Zhang, Z., Gold, A., and Surratt, J. D.: Examining the effects of anthropogenic emissions on isoprene-derived secondary organic aerosol formation during the 2013 Southern Oxidant and Aerosol Study (SOAS) at the Look Rock, Tennessee ground site, Atmospheric Chem. Phys., 15, 8871–8888, https://doi.org/10.5194/acp-15-8871-2015, 2015.

Hellén, H., Kouznetsov, R., Kraft, K., Seppälä, J., Vestenius, M., Jalkanen, J.-P., Laakso, L., and Hakola, H.: Shipping and algae emissions have a major impact on ambient air mixing ratios of non-methane hydrocarbons (NMHCs) and methanethiol on Utö Island in the Baltic Sea, Atmospheric Chem. Phys., 24, 4717–4731, https://doi.org/10.5194/acp-24-4717-2024, 2024.

Huybrechts, G., Narmon, M., and Van Mele, B.: The pyrolysis of CCl4 and C2Cl6 in the gas phase. Mechanistic modeling by thermodynamic and kinetic parameter estimation, Int. J. Chem. Kinet., 28, 27–36, https://doi.org/10.1002/(SICI)1097-4601(1996)28:1<27::AID-KIN4>3.0.CO;2-O, 1996.

Lim, K. P. and Michael, J. V.: Thermal decomposition of CH2Cl2, Symp. Int. Combust., 25, 809–816, https://doi.org/10.1016/S0082-0784(06)80714-9, 1994.

Schiemann, G., Immel, O., Rötger, H., and Schmidt, H.: Zur thermischen Zersetzung von Tetrachlorkohlenstoffdämpfen und ihre reaktionstechnische Auswertung, Z. Für Phys. Chem., 32, 137–153, 1962.

Shilov, A. and Sabirova, R.: The mechanism and the isotope effect of the primary stage in the thermal breakdown of chloroform, Dokl. Akad Nauk SSSR I14, 1058–61, 1957.

Shilov, A. and Sabirova, R.: Mechanism of the primary act of the thermal decomposition of methane derivatives, Zh Fiz Khim, 6, 1365–1373, 1959.

Tripathi, N., Sahu, L. K., Patel, K., Kumar, A., and Yadav, R.: Ambient air characteristics of biogenic volatile organic compounds at a tropical evergreen forest site in Central Western Ghats of India, J. Atmospheric Chem., 78, 139–159, https://doi.org/10.1007/s10874-021-09415-y, 2021.

Zhu, L. and Bozzelli, J. W.: Kinetics and mechanism for the thermal chlorination of chloroform in the gas phase: Inclusion of HCl elimination from CHCl3, Int. J. Chem. Kinet., 35, 647–660, https://doi.org/10.1002/kin.10159, 2003.

**In section 2.3.1 and on page 11: the CAPS instrument is know to be affected by changes in humidity. Is this a factor in your setup?**

The Teledyne T500U includes a drying assembly in its internal components and has a manufacturer recommended effective humidity use range between 0-95% relative humidity, which is now clarified in Sect. 2.3.1, lines 203-204. As evidenced by the good agreements between methods displayed in Figs. 3 and 5, we did not find the humidity ranges tested with CAPS to be a significant confounding factor for our experiments. Further, we found

the presence of the $NO_2$ artefact as measured by CAPS to be independent of humidity (ie., it was found under both dry and humid conditions).

**Minor comments**

**On line 149: could the the particle filter in front of the inlet affect the measurements in ambient conditions?**

We found no loss of $ClNO_2$ through the PFA Teflon filter. We tested the addition of $ClNO_2$ before the filter and after removal of the filter from the sampling line, and there was no significant difference between the detected signals. Should the filter collect substantial amounts of aerosol, it is possible heterogeneous chemistry could occur with the collected aerosol to artificially enhance observed $ClNO_2$ mixing ratios. We have added text to clarify this and further recommend regular replacement of filters where this technique is used, lines 157-160.

**On line 282: was the furnace heated for these experiments?**

We have clarified that the furnace was heated for these experiments line 327.

**On line 343: what changes in the CIMS inlet are you referring to here?**

The CIMS / TILDAS comparison experiments occurred during the spring / early summer 2024, while the ambient sampling experiments occurred in January 2025. In the time in between, maintenance was performed on the CIMS, including replacement of the reagent ion permeation source, repair of the reagent ion heaters to enable more stable output, and an overhaul of the overall experimental sampling configuration as reflected in Figs. 1b-c. Given that the CIMS signals were normalized as described in Sect. 2.3.2, it is unclear why any of these changes would have affected instrument sensitivity to $ClNO_2$; however, changes in sensitivity were also noted for compounds not relevant to this work that were calibrated before and after our experiments (e.g., $Cl_2$). We have clarified this (Lines 392-393).

**On page 3: either use the "g" and "(aq)" notation for all equations or use it for none of them.**

**On line 242: please add the bond dissociation energy of ClNO2 for easy comparison.**

**On line 273: delete "loss".**

**Figure 5: please add the Pearson's coefficients (on the figure or in the caption).**

We have addressed the above four comments as suggested.

**On line 274: delete "not".**

The sentence is correctly written as is.

**Anonymous Referee 2**

**General comments and questions:**

1. **The authors explore potential interferences from a number of Cl containing species including: $Cl_2$, HOCl, $ClONO_2$ and chloroamines. To do this the authors model the TD of these species within their inlet furnace using the Kineticus model. This evaluation is important given the interest in**

**using this technique for ambient ClNO$_2$ measurements, where multiple Cl containing species could dissociate under the operation conditions, contributing to Cl radical production (and HCl) and an overestimation of ClNO$_2$ mixing ratios. However, the authors do not mention potential interference from ClNO. ClNO has been reported in laboratory studies to form from the reaction of HCl with surfaces that have been exposed to NO2 (e.g., Raff et al., 2009) or reaction with HONO (Wingen et al., 2000), or through reaction of NO$_2$ with particulate chloride (Weis and Ewing, 1999), and hence could be of interest for ambient measurements. Although the reviewer is not aware of any ClNO measurements for the troposphere, it could in theory provide an additional interference for the quantification of ClNO$_2$, since it can dissociate to form Cl + NO (~160 kJ mol$^{-1}$).**

ClNO is a potential unaccounted-for source of thermolysable Cl as the reviewer states, and we thank them for pointing this out. While we are also unaware of any tropospheric observations of ClNO, it is an important consideration when using this technique for ambient measurements. We have performed an additional Kintecus model run (Fig. A2) that includes 1 ppbv of ClNO. While only 40% of ClNO dissociated at the same timescale as >99% of ClNO2, it does bias the HCl measurement high. We have added additional text (Lines 265-273) to make the reader aware. We have additionally verified in our CIMS data that no IClNO$^-$ was detected during ambient sampling nor during the generation of our ClNO$_2$ standard, and data for IClNO$^-$ ($m/z$ 191.871862) has been additionally added to our interference plot in Fig. A4 to show that it did not rise above baseline when our ClNO2 appeared to be affected by other interferences.

As ClNO was found to be only partially sensitive to thermal dissociation in the model, it is likely we could increase the selectivity of the method for ClNO2 by reducing the temperature of the furnace and increasing the residence time in the hot tube, thereby minimizing the potential thermal dissociation of ClNO, if found to be present.

2. **The authors briefly discuss the potential for chloroamines to act as a significant interference. The reviewer agrees with the authors that this is important to highlight. In order to further emphasize this, it would be helpful to expand upon the potential magnitude of this interference slightly. Specifically, where the greatest interferences from chloroamines might occur (and give an example of mixing ratios). For example, Wang et al., 2023. This would be useful for anyone considering adopting this technique for field measurements.**

We have added text (Lines 277-278) to indicate potential sources of chloramines (e.g., near pools, during periods when office buildings are being cleaned) where chloramine mixing ratios may be elevated and lead to potential interference. We have additionally provided an example of previously observed emissions (lines 417-420).

3. **Modelling was conducted using Kineticus to capture the TD profile of ClNO$_2$, formation of HCl and explore potential interferences through generation of Cl radicals from other chlorine containing species. I believe it would be useful for the reader to include a few additional details on the model (and initiation conditions and assumptions) for those who are not familiar with Kineticus. For example, it would be useful to provide context for why in Fig. 2b the number density**

**of ClNO$_2$ does not increase as the TD temperature decreases, whereas the OH and Cl continue to change during this period. In the current text, this is not clear.**

Kintecus is a 0D box model that can be used to explore gas phase chemistry. The initial conditions and model reactions are explicitly listed in Tables A1-A3 in the appendix. We have added additional references to these tables to make this clearer (Lines 233-237, 246). With regard to changing OH / Cl number densities in Fig. 2b, room temperature reactions will still be allowed to occur (such as OH / Cl + VOC or the reformation of ClNO$_2$ from Cl + NO$_2$) with unreacted OH and Cl. However, we believe that the y-axis scaling is not appropriate for this logarithmically scaled figure, as the magnitude of these species are several orders of magnitude below 1 molecule cm$^{-3}$ by the time the temperature is lowered in the model, and therefore not meaningful. We have changed the lower limit of the y-axis on Fig. 2b to 1 molecule cm$^{-3}$ to make this clearer for the reader and believe the modified figure more properly visualizes relevant processes.

4. **The authors show some 'proof of concept' ambient measurements of ClNO$_2$ which were conducted on Jan 13$^{th}$, 2025. According to the x-axis of Fig. 6 these measurements lasted for ~12 hours. The authors also state on lines 335-337 that during this period the throughput of ClNO$_2$ in the base-coated denuder (designed to remove HCl and other acids) varied from 55%-33%. Do the authors have a sense of how this trend would hold out over longer sampling periods, such as days or weeks? How would this impact corrections applied to ClNO$_2$ concentrations? Additionally, at what point might the authors expect breakthrough of acidic gases such as HCl, which may interfere with the ClNO$_2$ measurement. These would be important to consider when using this technique for longer field deployment.**

The freshly coated denuder was first characterized and allowed to sample from 10 Jan, and so the change in denuder influence on ClNO2 mixing ratios as presented will have occurred over approximately 4 days. We have clarified this point on line 386.

We believe the effect on ClNO$_2$ will be dependent both on the specific denuder used and the basic coating; as there will be slight differences to coatings as denuders are repeatedly refreshed, each use of a denuder should be characterized across individual experiments where this technique is deployed. This can be achieved through periodic injections of a ClNO2 standard into/bypassing the denuder to estimate ClNO2 loss factors. This loss characterization is now described by lines 171-175.

Regarding HCl breakthrough, our own anecdotal observations from separate experiments focusing on HCl sampling show no breakthrough over a time span of 1 week, after which the used denuder would be replaced with a freshly coated denuder. In light of this and the effects on ClNO2, users of this technique would likely need to replace a used denuder at least weekly. We have added this recommendation to the Conclusions section (lines 433-434).

**Specific comments:**

**Line 58- the authors refer to the instrument as a "N$_2$O$_5$-cavity ring down spectrometer" and reference Thaler et al. It is perhaps more appropriate to just say "TD-CRDS via quantification of NO$_2$ by absorption**

at 405 nm". The "$N_2O_5$" instrument in question also used a laser at 662 nm for quantification of $NO_3$ (from the TD of $N_2O_5$), which would not be used for $ClNO_2$.

We have adjusted the language to the reviewer's suggestion (lines 59-60, 67-68).

**Line 87- are the costs an order of magnitude lower? Would be useful for reader to give an approximate idea of how much cheaper this method is.**

We clarified our meaning to time-cost (line 88). As the authors were involved in the development of the commercial HCl-TILDAS instrument used, we do not feel it is appropriate to comment on the financial cost.

**Section 2.2- Where is the temperature of the quartz furnace measured?**

The temperature of the furnace was measured inside the furnace using the furnace's internal temperature sensors. We have added a statement clarifying this (line 132-133).

**Fig1 – it is not currently clear where there the additional 5 m of tubing for ambient measurements was added based on the schematic in Fig. 1c. Please consider adding a few measurements for key areas on the schematic.**

Roman numeral labels have been added to Fig. 1c at key areas, with approximate distances labelled in the Figure caption.

**Anonymous Referee 3**

**This manuscript describes a new method to quantify gas phase ClNO2 under ambient sampling conditions. The method couples a previously characterized approach for HCl measurement (TILDAS) with a thermal desorption approach that efficiently generates HCl from ClNO2. Potential biases and interferences are discussed, and good comparisons are made using existing techniques in lab and ambient conditions. The main advantages of the method appear to be lower cost and less demanding calibration needs than CIMS, the currently used technique. The main drawbacks are that HCl must first be removed from the sample air (necessitating a denuder and a correction for additional ClNO2 loss), and apparent interferents in the form of chloramines (or some other compound that is co-emitted). The authors acknowledge that further work is required to determine whether these interferents can be disentangled in a practical way for ambient measurements.**

**Overall the manuscript reads quite well and is organized in a logical way. I have only a few minor suggestions, noted below, but otherwise I find this work ready for final acceptance.**

**Figure 1. The readability of this could be improved- hard to read the small text, especially the gray color. Is this a vector image format? If not, that would be an improvement.**

We have increased the size of Fig 1 to improve its readability and removed the gray color.

**Figure 2b. It's pretty hard to differentiate the HCl and Cl colors on this figure. Consider adjusting at least one of those.**

The color of Cl has been adjusted to a dark gray.

**Line 246. It would be helpful to state known or estimated mixing ratios of chloramines in any contexts presented in existing literature here.**

We have included information of estimated mixing ratios of chloramines as observed in downtown Toronto by Wang et al. (2023) (lines 417-419).

**Line 369. Is there any other way besides temperature scans that this could be achieved?**

We believe temperature scans would be the simplest way to attempt to discriminate between the relative contributions of chloramines and $ClNO_2$ to the overall TILDAS. A hypothetical experiment would include separate pure sources of chloramines and $ClNO_2$, verified using an external method that can readily distinguish between these compounds, such as CIMS. From here the thermal dissociation profiles of the chloramines could be determined, and assessments could be made as to whether this method could be adjusted to selectively detect these compounds. In principle, a method for selectively scrubbing $ClNO_2$ from the sample stream while retaining chloramines could exist, but more experimental work would be required to investigate.